# Let a Neural Network Be Your Invariant

**Mirco Giacobbe**[*]
University of Birmingham, UK
Zeroth Research, UK

**Daniel Kroening**[*]
Amazon Web Services, USA

**Abhinandan Pal**[*]
University of Birmingham, UK
National Institute of Informatics Tokyo, Japan

**Michael Tautschnig**[*]
Amazon Web Services, USA
Queen Mary University of London, UK

## Abstract

Safety verification ensures that a system avoids undesired behaviour. Liveness complements safety, ensuring that the system also achieves its desired objectives. A complete specification of functional correctness must combine both safety and liveness. Proving with mathematical certainty that a system satisfies a safety property demands presenting an appropriate inductive invariant of the system, whereas proving liveness requires showing a measure of progress witnessed by a ranking function. Neural model checking has recently introduced a data-driven approach to the formal verification of reactive systems, albeit focusing on ranking functions and thus addressing liveness properties only. In this paper, we extend and generalise neural model checking to additionally encompass inductive invariants and thus safety properties as well. Given a system and a linear temporal logic specification of safety and liveness, our approach alternates a learning and a checking component towards the construction of a provably sound neural certificate. Our new method introduces a neural certificate architecture that jointly represents inductive invariants as proofs of safety, and ranking functions as proofs of liveness. Moreover, our new architecture is amenable to training using constraint solvers, accelerating prior neural model checking work otherwise based on gradient descent. We experimentally demonstrate that our method is orders of magnitude faster than the state-of-the-art model checkers on pure liveness and combined safety and liveness verification tasks written in SystemVerilog, while enabling the verification of richer properties than was previously possible for neural model checking.

## 1 Introduction

Model checking addresses the question of whether a reactive system $\mathcal{M}$ meets a specification of intended behaviour $\Phi$ [42]. Reactive systems are systems that continually respond to inputs and operate over infinite executions. Exemplars include hardware designs implementing processing units, communication and arbitration protocols, as well as digital controllers for automotive, aerospace, or medical applications, where safety is paramount. Safety properties require that *nothing bad ever happens*. A safety violation is a finite execution the last state of which falsifies $\Phi$ and the damage is irredeemable [77]. Safety properties are dual to liveness properties, which require that *something good eventually happens*. A liveness violation is evidenced by an infinite execution that falsifies $\Phi$, as any finite violating prefix could in principle still be extended to a compliant execution [77]. Every linear-time specification is the composition of a safety and a liveness property [8], and linear temporal logic (LTL) provides an expressive language for specifying linear-time properties of reactive systems.

---

[*]The authors are listed alphabetically.

39th Conference on Neural Information Processing Systems (NeurIPS 2025).

SystemVerilog Assertions (SVA) bring linear-time logic into commercial hardware design flows [56], and electronic design automation vendors have spent decades optimising symbolic model checkers for them. Testing and simulation alone cannot fully guarantee that a system meets its specifications, and explicitly enumerating every possible behaviour is impractical, except for the most trivial hardware designs. Symbolic model checking has made it feasible to handle large state spaces. Two symbolic paradigms dominate. Algorithms based on binary decision diagrams (BDDs) provide sound and complete procedures for LTL model checking, yet BDDs often explode in size when faced with arithmetic data paths ubiquitous in modern hardware design [26, 52]. Bounded model checking (BMC) algorithms enjoy the much more scalable combinatorial and arithmetic reasoning of SAT solvers, but only explore the state space up to a finite depth, sacrificing completeness over unbounded execution [17]. Incremental algorithms for the construction of exploration frontiers (IC3) bridge that gap in unbounded safety verification [20], but fall short on liveness.

Neural model checking is an automated approach that leverages *neural certificates* to verify reactive systems against LTL specifications, but prior work has exclusively focused on liveness properties [63]. This prior cognate work learns ranking functions represented as neural networks—which serve as proof of liveness—from random executions and then checks their validity using symbolic reasoning, thereby providing formal guarantees. However, specifications that characterise pure safety or combinations of safety and liveness additionally demand the construction of inductive invariants—which serve as proofs of safety. Prior work does not construct such invariants, leaving safety out of reach.

In this paper, we demonstrate that neural networks are an effective representation for inductive invariants as well. Our novel method leverages neural certificates to simultaneously represent ranking functions and inductive invariants. Given a hardware design $\mathcal{M}$ and a non-deterministic Büchi automaton $\mathcal{A}_{\neg\Phi}$ that recognises the violations to an LTL formula $\Phi$, we train a neural certificate on sample transitions of the synchronous composition $\mathcal{M} \parallel \mathcal{A}_{\neg\Phi}$. This network (i) classifies each state as inside or outside the invariant and (ii) ranks the states, within the invariant, to strictly decrease whenever an accepting state is encountered. We then pose a one-step BMC query to check if the neural certificate is true over the entire state space. On affirmation, this proves every accepting state is either (i) unreachable or (ii) visited at most finitely many times. This implies that no reachable behaviour of $\mathcal{M}$ satisfies $\neg\Phi$, thereby proving that every reachable behaviour of $\mathcal{M}$ satisfies $\Phi$.

We implemented our method using a new neural ranking function architecture that enables training using constraint solvers, in contrast to prior work that exclusively relies on gradient descent for training [63]. Our new architecture yields more compact representations and it is much more efficient and numerically stable than prior work. We extended the prior benchmark with standard safety and combined liveness and safety verification problems, and have compared our method with the state-of-the-art hardware model checkers. Measuring against the per-task fastest on pure-liveness, our method is faster on $66\,\%$ of tasks, $10\times$ faster on $46\,\%$, $10^3\times$ on $11\,\%$, and $10^4\times$ on $4\,\%$. On pure safety tasks, while not outperforming, we match the best symbolic tools: $80\,\%$ of instances complete for all approaches in under $1\,\mathrm{s}$. The picture changes dramatically for properties that demand *both* an inductive invariant and a ranking function: our method is faster than the leading model checkers on $61\,\%$ of tasks, running $10^2\times$ on $43\,\%$, $10^4\times$ on $27\,\%$, and $10^5\times$ on $6\,\%$.

Our contribution is threefold. First, by simultaneously representing inductive invariants and ranking functions using neural networks, we extend and generalise neural model checking to verify arbitrary LTL properties encompassing both safety and liveness. Second, by introducing a neural architecture amenable to training using constraint solvers, we achieve both superior runtime efficiency and numerical stability. Third, by evaluating our method across a 634-task benchmark suite, we demonstrate that our method outperforms the state of the art in automated formal verification.

## 2   Model Checking Safety and Liveness

We consider specifications $\Phi$ in LTL over a set of atomic propositions $\Pi$. LTL extends propositional logic with temporal operators: $\mathsf{X}\,\Phi$ meaning $\Phi$ holds at the next step, $\mathsf{F}\,\Phi$ meaning $\Phi$ eventually holds, $\mathsf{G}\,\Phi$ meaning $\Phi$ always holds, $\Phi_1\,\mathsf{U}\,\Phi_2$ meaning $\Phi_1$ holds at all times until $\Phi_2$ necessarily holds at some future step, and $\Phi_1\,\mathsf{W}\,\Phi_2$ meaning $\Phi_1$ holds at all time before $\Phi_2$ possibly occurs [102]. LTL naturally expresses safety, liveness, and their combinations for reactive systems. For $\Pi = \{\mathsf{a}, \mathsf{b}\}$, the formula $\mathsf{G}\,\mathsf{a}$ is a safety property: a single finite prefix whose last state violates $\mathsf{a}$ refutes it. By contrast, $\mathsf{F}\,\mathsf{b}$ is a liveness property: no finite execution suffices to disprove it, since $\mathsf{b}$ may still

occur later. The formula a U b combines both: its safety component forbids any prefix containing ¬a∧¬b before the first b, while its liveness component requires that b eventually occurs. Equivalently, a U b = a W b ∧ F b, where a W b is pure safety and F b is pure liveness.

We model a reactive system $\mathcal{M}$ over a finite variable set $X_{\mathcal{M}}$ as a state transition system by an initial condition $\text{Init}_{\mathcal{M}}$ and a sequential update relation $\text{Update}_{\mathcal{M}}$. The variables are split into inputs $\text{inp}\, X_{\mathcal{M}}$ and state-holding registers $\text{reg}\, X_{\mathcal{M}}$. $\text{Init}_{\mathcal{M}}$ is a first-order predicate over $\text{reg}\, X_{\mathcal{M}}$ and $\text{Update}_{\mathcal{M}}$ is a first-order predicate over $X_{\mathcal{M}} \cup \text{reg}\, X'_{\mathcal{M}}$, where the set of primed variables denotes the next-cycle values. A reactive system $\mathcal{M}$ induces a transition system with state space $S$ of valuations of $X_{\mathcal{M}}$, initial states $S_0 \subseteq S$ with $\boldsymbol{s} \in S_0 \iff \text{Init}_{\mathcal{M}}(\text{reg}\, \boldsymbol{s})$, and transition relation $\rightarrow_{\mathcal{M}} \subseteq S \times S$ with $\boldsymbol{s} \rightarrow_{\mathcal{M}} \boldsymbol{s}' \iff \text{Update}_{\mathcal{M}}(\boldsymbol{s}, \text{reg}\, \boldsymbol{s}')$. The executions of $\mathcal{M}$ are the infinite sequences $(\boldsymbol{s}_0, \boldsymbol{s}_1, \boldsymbol{s}_2 \ldots)$ where $\boldsymbol{s}_0 \in S_0$ and $\boldsymbol{s}_i \rightarrow_{\mathcal{M}} \boldsymbol{s}_{i+1}$ for all $i \in \mathbb{N}$.

We cast our model checking question as a language inclusion question. Given a system $\mathcal{M}$, we define as $\text{obs}\, X_{\mathcal{M}} \subseteq X_{\mathcal{M}}$ its set of observables, and in turn define the language $L_{\mathcal{M}}$ of $\mathcal{M}$ as the set of sequences $(\text{obs}\, \boldsymbol{s}_0, \text{obs}\, \boldsymbol{s}_1 \ldots)$—which we call the traces of $\mathcal{M}$—induced by the executions $(\boldsymbol{s}_0, \boldsymbol{s}_1 \ldots)$ of $\mathcal{M}$. Given an LTL formula $\Phi$ over atomic propositions $\Pi$ and the alphabet $\Sigma = 2^{\Pi}$ of valuations of $\Pi$, we define the language $L_{\Phi} \subseteq \Sigma^{\omega}$ of $\Phi$ as the set of all possible traces $(\sigma_0, \sigma_1 \ldots) \in \Sigma^{\omega}$ satisfying $\Phi$. Provided that $\Pi = \text{obs}\, X_{\mathcal{M}}$ (which we require to be Boolean), the model checking question asks whether $L_{\mathcal{M}} \subseteq L_{\Phi}$, i.e., all traces of $\mathcal{M}$ satisfy $\Phi$. This is equivalent to the language emptiness question $L_{\mathcal{M}} \cap L_{\neg\Phi} = \emptyset$, i.e., no trace of $\mathcal{M}$ violates $\Phi$ [11, 42].

As is standard in automata-theoretic verification, we reduce the language emptiness question to an equivalent *fair emptiness* question. A non-deterministic Büchi automaton $\mathcal{A}$ consists of a finite set of states $Q$, an initial state $q_0 \in Q$, input alphabet $\Sigma$, transition relation $\delta \subseteq Q \times \Sigma \times Q$, and a set of fair states $F \subseteq Q$ [8, 39, 125]. This can be viewed as a reactive system with a single variable $X_{\mathcal{A}} = \{\mathsf{q}\}$ over $Q$, where $\text{Init}_{\mathcal{A}}(q) \iff q = q_0$ and $\text{Update}_{\mathcal{A}}(\sigma, q, q') \iff (q, \sigma, q') \in \delta$. We define the fair language $L^{\mathsf{f}}_{\mathcal{A}}$ of $\mathcal{A}$ as the set of traces of $\mathcal{A}$ whose respective executions visit $F$ infinitely often. We rely on the fact that every LTL formula $\Phi$ admits a non-deterministic Büchi automaton $\mathcal{A}_{\Phi}$ for which $L^{\mathsf{f}}_{\mathcal{A}_{\Phi}} = L_{\Phi}$. Our reduction to fair emptiness thus constructs an automaton $\mathcal{A}_{\neg\Phi}$ and forms the synchronous product $\mathcal{M} \parallel \mathcal{A}_{\neg\Phi}$ between $\mathcal{M}$ and $A_{\neg\Phi}$ with the fair states defined as $S \times F$. We refer the reader to the literature for a full formal definition of the synchronous product [9, 11, 42]; the key property to note is that $L^{\mathsf{f}}_{\mathcal{M}\parallel\mathcal{A}_{\neg\Phi}} = L_{\mathcal{M}} \cap L^{\mathsf{f}}_{\mathcal{A}_{\Phi}}$. Hence, our language emptiness question and, in turn, our model checking question amount to deciding the fair emptiness question $L^{\mathsf{f}}_{\mathcal{M}\parallel\mathcal{A}_{\neg\Phi}} = \emptyset$.

We answer the fair emptiness question by presenting a certificate consisting of two components: (i) an inductive invariant defined as a set $I \subseteq \text{reg}\, S \times Q$ that captures (or over-approximates) the reachable states of $\mathcal{M} \parallel \mathcal{A}_{\neg\Phi}$, and (ii) a ranking function $V : I \rightarrow \mathbb{W}$ over a well-founded relation $(\mathbb{W}, \prec)$ that assigns ranks to all states within the invariant that prove that all reachable executions visit fair states at most finitely many times. Concretely, for all $\boldsymbol{s}, \boldsymbol{s}' \in S$ and $q, q' \in Q$,

$$\boldsymbol{s} \in S_0 \qquad\qquad\qquad\qquad\qquad\qquad \implies (\text{reg}\, \boldsymbol{s}, q_0) \in I, \tag{1}$$

$$(\boldsymbol{s}, q) \rightarrow_{\mathcal{M}\parallel\mathcal{A}_{\neg\Phi}} (\boldsymbol{s}', q') \ \wedge\ (\text{reg}\, \boldsymbol{s}, q) \in I \qquad \implies (\text{reg}\, \boldsymbol{s}', q') \in I, \tag{2}$$

$$(\boldsymbol{s}, q) \rightarrow_{\mathcal{M}\parallel\mathcal{A}_{\neg\Phi}} (\boldsymbol{s}', q') \ \wedge\ (\text{reg}\, \boldsymbol{s}, q) \in I \qquad \implies V(\text{reg}\, \boldsymbol{s}, q) \succeq V(\text{reg}\, \boldsymbol{s}', q'), \tag{3}$$

$$(\boldsymbol{s}, q) \rightarrow_{\mathcal{M}\parallel\mathcal{A}_{\neg\Phi}} (\boldsymbol{s}', q') \ \wedge\ (\text{reg}\, \boldsymbol{s}, q) \in I \ \wedge\ q \in F \quad \implies V(\text{reg}\, \boldsymbol{s}, q) \succ V(\text{reg}\, \boldsymbol{s}', q'). \tag{4}$$

Clauses (1) and (2) state that initial states lie in $I$ and that $I$ is closed under transitions, so $I$ forms an over-approximation of the set of reachable states. Clauses (3) and (4) require that, along every transition whose source is in $I$, the rank $V$ never increases, and it decreases strictly whenever the source state is fair ($q \in F$). Because $(\mathbb{W}, \prec)$ is well founded, only finitely many strict decreases are possible; hence, fair states must be visited at most finitely many times. This implies that all executions are unfair, i.e., $L^{\mathsf{f}}_{\mathcal{M}\parallel\mathcal{A}_{\neg\Phi}} = \emptyset$, which is equivalent to saying all traces of $\mathcal{M}$ satisfy $\Phi$ [125].

Figure 1 illustrates our workflow on an example. Figure 1a gives a SystemVerilog module $\mathcal{M}$ that satisfies the LTL property $\Phi = \mathsf{a\ U\ b}$. Figure 1b gives a Büchi automaton $\mathcal{A}_{\neg\Phi}$ recognising $\neg\Phi$. Figure 1c depicts the state space of the synchronous product $\mathcal{M} \parallel \mathcal{A}_{\neg\Phi}$: product states are arranged on a two-dimensional grid, each state corresponding to the pair of a state $s_i$ of $\mathcal{M}$ (where $i$ is the value of the register c) and a state $q_0$, $q_1$, or $q_2$ of $\mathcal{A}_{\neg\Phi}$. The sole initial state is $(s_0, q_0)$, and the dotted region indicates the inductive invariant $I$. Each state in $I$ indicates in its upper-right corner the respective rank, taken from $\mathbb{N}$. Along each transition in $I$, the rank strictly decreases when leaving a fair state and never increases otherwise, ensuring that the sole fair cycle within $I$ terminates. Notably,

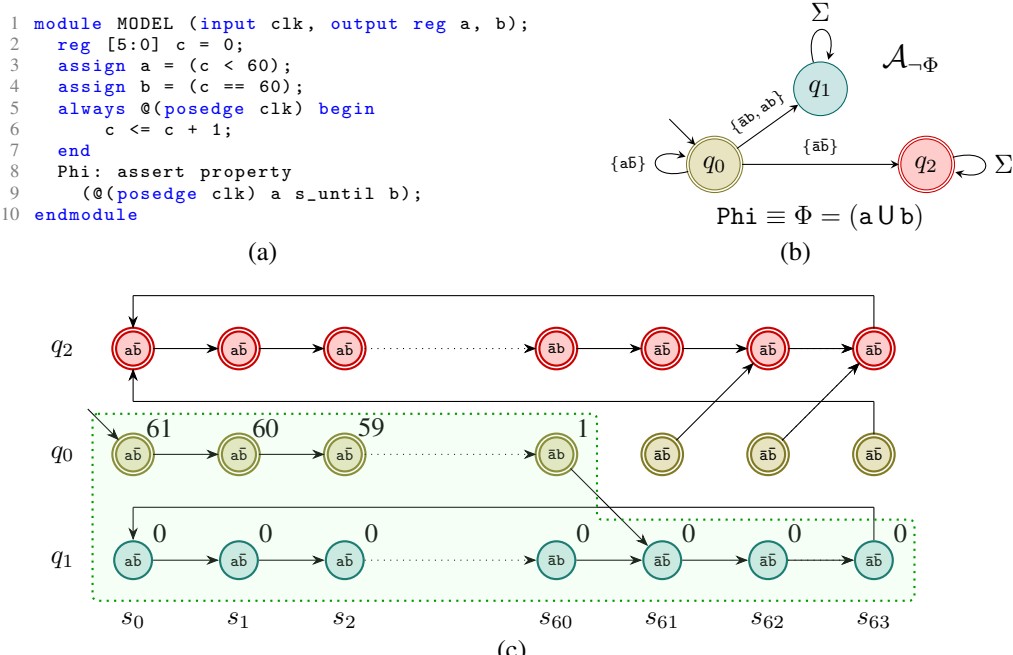

```
1  module MODEL (input clk, output reg a, b);
2    reg [5:0] c = 0;
3    assign a = (c < 60);
4    assign b = (c == 60);
5    always @(posedge clk) begin
6        c <= c + 1;
7    end
8    Phi: assert property
9      (@(posedge clk) a s_until b);
10 endmodule
```

(a)

$\text{Phi} \equiv \Phi = (a \cup b)$

(b)

(c)

Figure 1: Model checking safety and liveness on an illustrative example

the system has a non-terminating fair cycle (e.g., the cycle on $q_2$), but this is unreachable and excluded by $I$. In general, invariants may include unreachable states, provided none can lead to a fair cycle.

## 3  Neural Partially-Ranking Functions

We reduce the fair emptiness problem to a machine learning task by (i) modelling the invariant $I$ as a *classifier* over-approximating the reachable states, and (ii) modelling the ranking function $V$ as a *regressor*. In this section, we present our learn-check workflow which, upon termination of our procedure, formally ensures that these two components jointly satisfy conditions (1)–(4). Crucially, we introduce the *neural partially-ranking function*, which combines the definition of inductive invariant and ranking function into a single neural network.

We define a neural partially-ranking function over discrete parameter space $\Theta$ as the function

$$\bar{V} : \operatorname{reg} S \times \Theta \to \mathbb{Z} \tag{5}$$

Our objective is to learn an upper bound $\kappa \in \mathbb{Z}$ and a set of parameters $\{\theta_q \in \Theta\}_{q \in Q}$, such that for all $s, s' \in S$ and $q, q' \in Q$, the following two conditions hold:

$$s \in S_0 \implies \kappa \geq \bar{V}(\operatorname{reg} s; \theta_{q_0}), \tag{6}$$

$$(s, q) \to_{\mathcal{M} \| \mathcal{A}_{\neg \phi}} (s', q') \wedge \kappa \geq \bar{V}(\operatorname{reg} s; \theta_q) \implies \bar{V}(\operatorname{reg} s; \theta_q) \geq \bar{V}(\operatorname{reg} s'; \theta_{q'}) + \mathbf{1}_F(q). \tag{7}$$

where $\mathbf{1}_F(q) = 1$ if $q \in F$ else 0. The trainable network parameters are in $\Theta$ and are distinct for each automaton state $q \in Q$, while $\kappa$ is global. The *initiation condition* of Eq. (6) ensures that all initial states receive a rank bounded above by $\kappa$. The *ranking condition* of Eq. (7) ensures that states bounded above by $\kappa$ are inductively assigned a rank. Ultimately, a solution $\boldsymbol{\kappa}$ and $\{\boldsymbol{\theta}_q\}_{q \in Q}$ induces an invariant $I$ and ranking function $V$ satisfying the clauses (1)–(4), respectively defined as follows:

$$I = \left\{ (\operatorname{reg} s, q) : \boldsymbol{\kappa} \geq \bar{V}(\operatorname{reg} s; \boldsymbol{\theta}_q) \right\}, \qquad V(\operatorname{reg} s, q) = \bar{V}(\operatorname{reg} s; \boldsymbol{\theta}_q) \text{ if } (\operatorname{reg} s, q) \in I. \tag{8}$$

Since the hardware state space $S$ and $Q$ are finite, the image $\bar{\mathbb{W}} = \{\bar{V}(\operatorname{reg} s; \boldsymbol{\theta}_q) : s \in S, q \in Q\}$ of $\bar{V}$ for a constant parameter $\{\boldsymbol{\theta}_q\}_{q \in Q}$ is finite as well, having a least element under the strict inequality $<$ over the integers. This makes $(\bar{\mathbb{W}}, <)$ well founded, having no infinite descending chain.

We train $\bar{V}$ in a counterexample-guided inductive synthesis loop as illustrated in Figure 2a [1, 117]. Our procedure incrementally constructs two datasets $D_{\mathsf{init}}$ and $D_{\mathsf{trans}}$ to train $\bar{V}$ to respectively satisfy

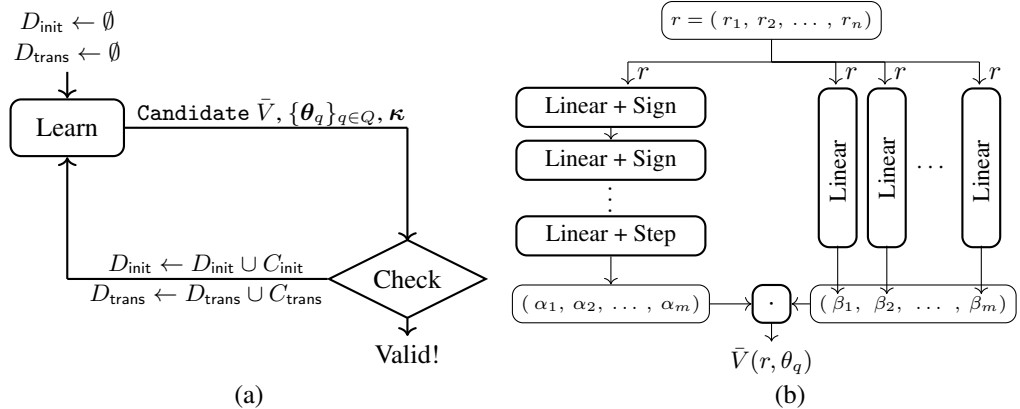

Figure 2: Our learn-check workflow (a) and our architecture for neural partially-ranking functions (b)

conditions (6) and (7). The dataset $D_{\text{init}}$ contains sample initial register values $\text{reg}\,\boldsymbol{s}$ of $\mathcal{M}$ where $\boldsymbol{s} \in S_0$, and the dataset $D_{\text{trans}}$ contains sample transitions of $\mathcal{M} \parallel \mathcal{A}_{\neg\phi}$ in the form of quadruples $(\text{reg}\,\boldsymbol{s}, \boldsymbol{q}, \text{reg}\,\boldsymbol{s}', \boldsymbol{q}')$ where $(\boldsymbol{s}, \boldsymbol{q}) \rightarrow_{\mathcal{M}\parallel\mathcal{A}_{\neg\phi}} (\boldsymbol{s}', \boldsymbol{q}')$. Initially, we assume that $D_{\text{init}}$ and $D_{\text{trans}}$ are empty. Our learning component computes a set of parameters $\{\boldsymbol{\theta}_q\}_{q\in Q}$ to hold over the set of samples $D_{\text{init}}$ and $D_{\text{trans}}$. Our checking component formally verifies whether conditions (6) and (7) hold over the entire state space. If the checker confirms the conditions, then the procedure halts successfully. In the converse case, it provides sets of counterexamples $C_{\text{init}}$ and $C_{\text{trans}}$, which are respectively added to $D_{\text{init}}$ and $D_{\text{trans}}$, and learning and checking are repeated in a loop.

**Learning**   Our learning phase computes a set of parameters for which our initiation and ranking conditions are valid across all samples in the dataset. More precisely, enforcing our initiation condition amounts to satisfying the following constraint:

$$\text{LearnInit}(\theta, \kappa) = \bigwedge_{\boldsymbol{r}\in D_{\text{init}}} \left(\kappa \geq \bar{V}\big(\boldsymbol{r}; \theta_{q_0}\big)\right) \tag{9}$$

Similarly, satisfying our ranking condition amounts to satisfying the following constraint:

$$\text{LearnRank}(\theta, \kappa) = \bigwedge_{(\boldsymbol{r}, \boldsymbol{q}, \boldsymbol{r}', \boldsymbol{q}')\in D_{\text{trans}}} \left(\kappa \geq \bar{V}(\boldsymbol{r}; \theta_{\boldsymbol{q}}) \Longrightarrow \bar{V}(\boldsymbol{r}; \theta_{\boldsymbol{q}}) \geq \bar{V}(\boldsymbol{r}'; \theta_{\boldsymbol{q}'}) + \mathbf{1}_F(\boldsymbol{q})\right) \tag{10}$$

Overall, our learning phase is tasked to solve the following question which, as we show in Section 4, we cast to a constraint satisfaction problem:

$$\exists \theta, \kappa \colon \text{LearnInit}(\theta, \kappa) \wedge \text{LearnRank}(\theta, \kappa) \tag{11}$$

If a valid set of parameters exists, then this is passed on to the checking phase. If a valid set of parameters does not exist, this indicates a larger network is needed or no solution exists at all. In this case, our strategy is to progressively add a hidden layer or widen it and retry until a preset limit.

**Checking**   Our checking phase determines the presence or the absence of counterexamples to our initiation and ranking conditions for the candidate parameters $\boldsymbol{\kappa}, \{\boldsymbol{\theta}_q\}_{q\in Q}$. More precisely, a counterexample for the initiation condition must satisfy the following constraint:

$$\text{CheckInit}(s) = \text{Init}_{\mathcal{M}\parallel\mathcal{A}_{\neg\Phi}}(s, q_0) \wedge \boldsymbol{\kappa} < \bar{V}(\text{reg}\,s; \boldsymbol{\theta}_{q_0}) \tag{12}$$

Similarly, a counterexample for the ranking condition must satisfy the following constraint:

$$\text{CheckRank}(s, q, r', q') = \text{Update}_{\mathcal{M}\parallel\mathcal{A}_{\neg\Phi}}(s, q, r', q') \wedge$$
$$\boldsymbol{\kappa} \geq \bar{V}(\text{reg}\,s; \boldsymbol{\theta}_q) \wedge \bar{V}(\text{reg}\,s; \boldsymbol{\theta}_q) < \bar{V}(r'; \boldsymbol{\theta}_{q'}) + \mathbf{1}_F(q) \tag{13}$$

Overall, our checking phase asks whether the following statement holds:

$$\exists s \colon \text{CheckInit}(s) \quad \vee \quad \exists s, q, r', q' \colon \text{CheckRank}(s, q, r', q') \tag{14}$$

We check the initialisation and ranking conditions independently. If both queries are unsatisfiable, then we conclude that $\bar{V}$ is a valid neural partially-ranking function. If either or both are satisfiable,

then the respective assignments constitute counterexamples. To produce multiple counterexamples $C_{\text{init}}$ or $C_{\text{trans}}$ in each iteration, we check (13) over explicitly enumerated automaton pairs $(q, q')$ independently. This accelerates convergence, and gives equal emphasis to each transition of $\mathcal{A}_{\neg\Phi}$.

Our counterexample-guided inductive synthesis procedure captures task-specific edge cases that yield succinct datasets. This is in contrast to prior work that benefits from random initialisation of the datasets before the first iteration [63]. Moreover, as we show in Section 4, we further optimise our procedure by proposing a specialised architecture that is amenable not only to checking but also to learning using efficient and numerically stable constraint satisfaction solvers.

## 4 Learning Neural Partially-Ranking Functions Using MILP Solvers

We present a specialised architecture for neural partially-ranking functions amenable to training using constraint satisfaction solvers. Our architecture is illustrated in Figure 2b. Given a vector of word-level register values $r = (r_1, \ldots, r_n)$ with $n = |\operatorname{reg} X_{\mathcal{M}}|$, our architecture feeds $r$ into two branches. The left component $\alpha^{(r,q)}$ consists of $L$ fully connected layers $(W_1^{(q)}, b_1^{(q)}, \ldots, W_L^{(q)}, b_L^{(q)})$ with $N_1, \ldots, N_L$ neurons, respectively, sign activation functions on the hidden layers, and step function on the output layer featuring $m = N_L$ neurons. The right component consists of $m$ linear functions with $n$ inputs each: $\beta^{(r,q)} = A^{(q)}r + c^{(q)}$. The left component $\alpha$ acts as a *mask* that selects or excludes the linear functions of the right component $\beta$; in other words, $\bar{V}(r, \theta_q) = \alpha^{(r,q)} \cdot \beta^{(r,q)}$.

Every application of arguments $(r, \theta_q)$ to $\bar{V}$ in Eqs. (9)–(11) requires encoding. We build upon the result for which feed-forward neural networks with sign activation functions are amenable to training with mixed-integer linear programming (MILP) solvers [67, 121]. Henceforth, we use $i, j$ to index individual elements of linear components, specifically $i$ for the inputs elements and $j$ for output elements, and we use $l$ to index layers of the network. A single hyperparameter $P \in \mathbb{N}_{>0}$ bounds the magnitude of all decision variables in the encoding. Decision variables shared across multiple occurrences of $r$ and associated to a common $q$ are denoted as $w_{i\ell j}^{(q)}, b_{\ell j}^{(q)}, a_{ij}^{(q)}, c_j^{(q)} \in [-P, P]$ to respectively refer to the elements of $W_l^{(q)}, b_l^{(q)}, A^{(q)}$ and $c^{(q)}$; the threshold $\kappa \in [-P, P]$ is also a common decision variable in the learning phase. Decision variables associated to specific occurrences of $r$ and $q$ are $u_{\ell j}^{(r,q)} \in \{0, 1\}$ to indicate the activation status of the corresponding neurons, $z_{i\ell j}^{(r,q)}$ to represent neuron-weight products where $z_{i1j}^{(r,q)} \in [-PM_r, PM_r]$ with $M_r = \max(|r_1|, \ldots, |r_n|)$ and $z_{i\ell j}^{(r,q)} \in [-P, P]$ with $l \geq 2$, and finally $v_j^{(r,q)} \in [-PM_r, PM_r]$ to represent the overall output of the network. First, we encode the relationship between $z, r$ and $w$ at layer 1 as

$$z_{i1j}^{(r,q)} = r_i w_{i1j}^{(q)} \tag{15}$$

Then, for $l = 1, \ldots, L$, we encode the relationship between $u, z$ and $b$ at layer $l$ (where $\epsilon > 0$) as

$$\left(u_{\ell j}^{(r,q)} = 1 \Rightarrow b_{\ell j} + \sum_{i \in N_{\ell-1}} z_{i\ell j}^{(r,q)} \geq \epsilon\right) \wedge \left(u_{\ell j}^{(r,q)} = 0 \Rightarrow b_{\ell j} + \sum_{i \in N_{\ell-1}} z_{i\ell j}^{(r,q)} \leq -\epsilon\right) \tag{16}$$

For $l = 2, \ldots, L$, we encode the relationship between $u$ at layer $l-1$ and $z$ and $w$ at layer $l$ as

$$
\begin{aligned}
&(z_{i\ell j}^{(r,q)} - w_{i\ell j}^{(q)} + 2P\, u_{(\ell-1)i}^{(r,q)} \leq 2P) \wedge (z_{i\ell j}^{(r,q)} + w_{i\ell j}^{(q)} - 2P\, u_{(\ell-1)i}^{(r,q)} \leq 0) \wedge \\
&(z_{i\ell j}^{(r,q)} - w_{i\ell j}^{(q)} - 2P\, u_{(\ell-1)i}^{(r,q)} \geq -2P) \wedge (z_{i\ell j}^{(r,q)} + w_{i\ell j}^{(q)} + 2P\, u_{(\ell-1)i}^{(r,q)} \geq 0)
\end{aligned} \tag{17}
$$

which represents $z_{i\ell j}^{(r,q)} = w_{i\ell j}$ if $u_{(\ell-1)j}^{(r,q)} = 1$, and $z_{i\ell j}^{(r,q)} = -w_{i\ell j}$ otherwise. Finally, we encode the relationship between $v, a, c$ and $u$ at layer $L$ as

$$\left(u_{Lj}^{(r,q)} = 1 \Rightarrow v_j^{(r,q)} = c_j^{(q)} + \sum_{i \in n} r_i a_{ij}^{(q)}\right) \wedge \left(u_{Lj}^{(r,q)} = 0 \Rightarrow v_j^{(r,q)} = 0\right) \tag{18}$$

As a result, each occurrence of $\bar{V}(r, \theta_q)$ in Eqs. (9)–(11) corresponds to the linear term $\sum_{i \in m} v_i^{(r,q)}$.

Notably, a naïve encoding that represents $z_{i\ell j}^{(r,q)}$ as a product between a neuron variable and $w_{i\ell j}^{(q)}$ would induce bilinear constraints; our encoding avoids this and produces an equivalent encoding with linear constraints only. This enables using efficient MILP (and SMT) solvers in the learning phase.

Essentially, our architecture represents a piecewise function with $2^m$ configurations, given by $\alpha \in \{0, 1\}^m$, each of which induces a specific summation of linear functions, given by $\beta \in \mathbb{Z}^m$, in the spirit of piecewise-linear ranking functions [76, 123, 124]. In our implementation, we restrict all decision variables to the *integer* type, which results in neural networks that are naturally quantised.

Table 1: Tasks completed per tool, per design; in bold the max per design, per spec-type

| | LS | LCD | Tmcp | i2cS | 7-Seg | PWM | VGA | UARTt | Delay | Gray | Blink | Total |
|---|---|---|---|---|---|---|---|---|---|---|---|---|
| Pure Safety | | | | | | | | | | | | |
| Tasks | 16 | 28 | 17 | 20 | 15 | 12 | 20 | 10 | 32 | 11 | 25 | 206 |
| Our | **16** | **28** | **17** | 4 | **15** | **12** | **20** | **10** | **32** | **11** | **25** | 190 |
| ABC | **16** | **28** | **17** | 20 | **15** | **12** | **20** | **10** | **32** | **11** | **25** | 206 |
| nuXmv | **16** | **28** | **17** | 20 | **15** | **12** | **20** | **10** | **32** | **11** | **25** | 206 |
| rIC3 | **16** | **28** | **17** | 20 | **15** | 2 | **20** | **10** | **32** | **11** | **25** | 196 |
| Pure Liveness | | | | | | | | | | | | |
| Tasks | 16 | 14 | 17 | 20 | 30 | 12 | 10 | 10 | 32 | 33 | 25 | 219 |
| Our | 0 | **14** | **17** | **20** | **30** | **12** | **10** | **10** | **32** | **33** | 14 | 192 |
| ABC | 3 | 5 | 9 | 4 | 10 | 3 | 5 | **10** | 8 | 16 | 5 | 78 |
| nuXmv | 12 | 13 | **17** | 16 | 28 | 7 | 4 | **10** | **32** | **33** | 8 | 180 |
| rIC3 | 7 | 7 | 13 | 7 | 17 | 4 | 6 | **10** | 14 | 19 | 6 | 110 |
| NMC'24 | **15** | 3 | 10 | **20** | **30** | 10 | 9 | **10** | **32** | **33** | 0 | 172 |
| Safety + Liveness | | | | | | | | | | | | |
| Tasks | 32 | 14 | 17 | 20 | 15 | 12 | 10 | 10 | 32 | 22 | 25 | 209 |
| Our | **32** | **14** | **17** | 3 | 9 | **12** | **10** | **10** | **32** | **22** | 11 | 172 |
| ABC | 7 | 12 | 11 | **20** | 5 | 3 | **10** | 7 | 6 | 11 | 4 | 96 |
| nuXmv | 11 | 12 | 14 | 16 | **15** | 6 | **10** | **10** | 26 | 14 | **20** | 154 |
| rIC3 | 12 | **14** | 14 | **20** | 8 | 4 | **10** | 9 | 18 | 14 | 6 | 129 |

## 5  Experimental Evaluation

**Implementation**  We implemented a prototype using a modular `Python` 3.12 pipeline [2]. Given an LTL formula $\Phi$, we obtained $\mathcal{A}_{\neg\Phi}$ with `Spot` 2.11.6 [51], converted SystemVerilog to bit-vector SMT using `EBMC` 5.6 [91], performed validity queries and computed counterexamples using `Bitwuzla` 0.7.0 [96], and trained a neural partially-ranking function using the MILP solver `Gurobi` 12.0.1 [65]. Our method relies on a single global hyperparameter configuration, besides the integer bound $P$ (see Section 4). We chose $P$ from a finite set of configurations $\{1, 5, 10, \lfloor M/10 \rfloor, \lfloor M/2 \rfloor, M, M{+}1, 2M\}$, where $M$ is the largest value any register can take in the hardware design; this renders the bounds for $z_{i1j}$ and $v_j$ quadratic in $M$ except when $P = 1$, 5, or 10. We configured the architecture to start with a linear model (per $q \in \mathcal{A}_{\neg\phi}$) and add one hidden layer with a single neuron (left branch of Figure 2a) upon the first failure of the learning phase, subsequently increasing its width by one neuron upon subsequent failures, up to 5 attempts.

**Experimental Setup**  Building on the ten parameterised SystemVerilog designs and 194 pure-liveness tasks of prior work [63], we add an eleventh design and, for all eleven, contribute additional pure-safety and combined safety–liveness specifications on each task, yielding a 634-task suite that retains the original diversity (Appendix A). We compared against (i) the gradient-based neural model checking approach, labelled `NMC'24` (liveness only) [63], and (ii) the winners of recent HWMCC editions [107]: `nuXmv` 2.1.0 [28], `rIC3` 1.5.1 [119], and `ABC`/Super-Prove [24]. For `nuXmv` we translated each design with its assertions to SMV using `EBMC`; then, we converted SMV to AIG for `ABC` and `rIC3` with `AIGER` [18], yielding a fully automated flow. All experiments ran on an AWS m6a.4xlarge instance (16 vCPUs, 64 GB RAM, Ubuntu 24.04). Each task had a 5 h timeout per tool; in aggregate, the experiments consumed 9083 h ($\approx$378 d) of compute time.

**Overall Efficacy**  Table 1 presents all solved instances by property class. For *pure-safety*—historically emphasised in HWMCC [16, 107]—symbolic tools clear $\geq 95\%$ of tasks, and we remain close to decades-mature tools at 92%. For *pure-liveness*, performance spreads: `nuXmv` 82%, `ABC` 36%, `rIC3` 50%, `NMC'24` 79%, while we complete 88%. On *safety–plus-liveness* properties, the gap widens, `nuXmv` solves 74%, `ABC` 46%, `rIC3` 61%, and our approach reaches 82%.

**Comparison on Pure Safety**  We generally perform on par with leading symbolic checkers with decades of optimisation, as Figure 3a indicates, while being faster than `NuXmv` on 17%, `rIC3` on

---

[2]https://github.com/aiverification/neuralmc

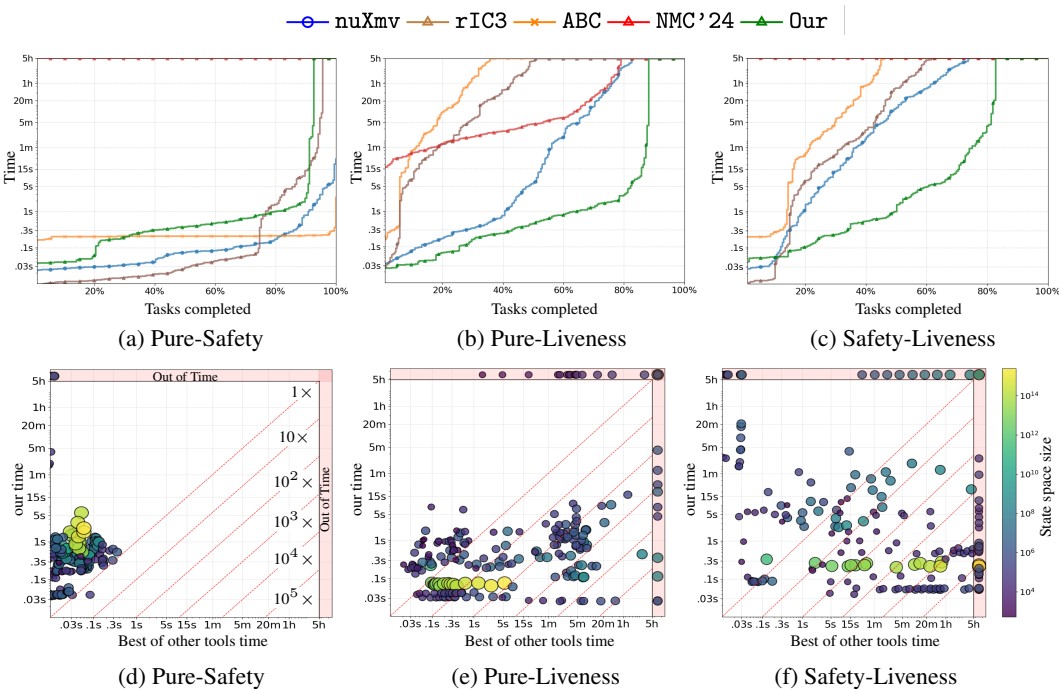

Figure 3: Runtime comparison with the state of the art (all times are in *log scale*)

$25\,\%$, and ABC on $32\,\%$ of safety tasks. Figure 3d plots each individual task by time, with the per-task fastest symbolic solver on the $x$-axis and our method on the $y$-axis; point size and brightness reflect state-space size. While symbolic tools often lead on individual instances, in $80\,\%$ of tasks both finish within $1\,\mathrm{s}$, indicating that a our method can meaningfully match a portfolio of symbolic safety checkers, besides being the first neural model checking framework to support general safety.

**Comparison on Pure Liveness** Figure 3b indicates that our method is consistently faster than the others. Within $1\,\mathrm{s}$ we solve $24\,\%$ more tasks than nuXmv, $59\,\%$ more tasks than rIC3, $59\,\%$ more than ABC, and $64\,\%$ more than NMC'24; at $1\,\mathrm{min}$ the margins widen to $31\,\%$, $68\,\%$, $76\,\%$, $71\,\%$, and they still measure $12\,\%$, $45\,\%$, $59\,\%$, $13\,\%$ after $1\,\mathrm{h}$. The percentage of tasks completed in $5\,\mathrm{h}$ per task by nuXmv, NMC'24, rIC3, and ABC are completed in under $7\,\mathrm{s}$, $3\,\mathrm{s}$, $0.6\,\mathrm{s}$, $0.3\,\mathrm{s}$ by our method. Figure 3d shows the per-task runtime gains against the fastest competing method per task: our method is faster on $66\,\%$ of tasks, at least $10\times$ faster on $46\,\%$, $10^2\times$ faster on $29\,\%$, $10^3\times$ faster on $11\,\%$, $10^4\times$ faster on $4\,\%$, and $10^5\times$ faster on $1\,\%$ with respect to the whole state of the art. On a specific comparison with NMC'24 we obtain speedups of at least $10\times$ on $88\,\%$, $10^2\times$ on $79\,\%$, $10^3\times$ on $45\,\%$, $10^4\times$ on $18\,\%$, and $10^5\times$ on $5\,\%$. Finally overall, $5\,\%$ of tasks timeout for all tools except ours.

**Comparison on Combined Safety and Liveness** Figure 3c indicates that the performance gap between our method and the alternatives further widens on safety-liveness tasks. Within $1\,\mathrm{s}$ our approach completes $31\,\%$ more tasks than nuXmv, $34\,\%$ more than rIC3 and $36\,\%$ more than ABC; after $1\,\mathrm{min}$ the gaps grow to $38\,\%$, $43\,\%$, $54\,\%$, and persist at $20\,\%$, $31\,\%$, $41\,\%$ after $1\,\mathrm{h}$. Benchmarks that exhaust nuXmv, ABC or rIC3 in $5\,\mathrm{h}$ finish in $56\,\mathrm{s}$, $6\,\mathrm{s}$, $0.8\,\mathrm{s}$ per task with our approach. Figure 3f shows that we are faster on $61\,\%$, $10\times$ faster on $52\,\%$, $10^2\times$ faster on $43\,\%$, $10^3\times$ faster on $36\,\%$, $10^4\times$ faster on $27\,\%$, and $10^5\times$ faster on $6\,\%$ of the tasks; all tools, besides ours, hit the $5\,\mathrm{h}$ timeout on $18\,\%$ of the task suite, bringing arbitrary LTL verification to runtime parity with pure-safety.

**Neural Certificates Expressivity** In Figure 4a, Auto is our default configuration that starts with linear for each state $q$ of $\mathcal{A}_{\neg\Phi}$ and escalates to neural models only if linear is infeasible; Only Linear is the ablation that *exits* immediately when the linear model fails. The linear model completes $48\,\%$ of tasks within $9\,\mathrm{s}$ per task and makes no progress thereafter; by contrast, our default finishes $87\,\%$ overall. Since the default always tries the linear model first, it never trails the ablation; when linear

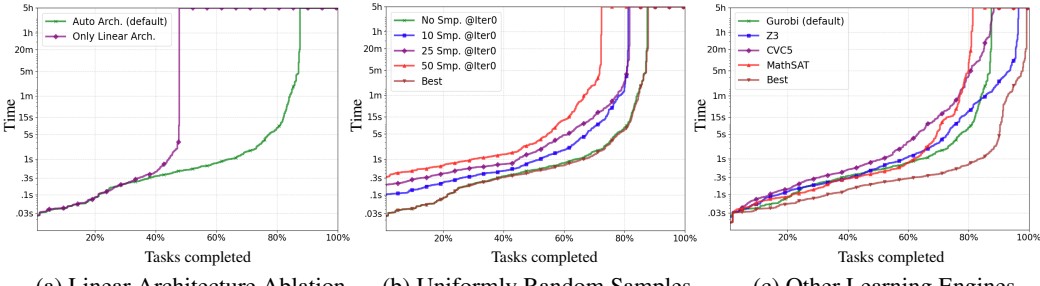

(a) Linear Architecture Ablation (b) Uniformly Random Samples (c) Other Learning Engines

Figure 4: Runtime analysis of our approach (all times are in *log scale*)

fails, automatic widening provides the expressivity needed to solve a lot more instances, showing that purely affine certificates are insufficient while our neural-certificate architecture succeeds.

**Comparison with Random Dataset Generation**   Our iterative procedure presented in Figure 2a starts with empty datasets $D_{\text{init}}$ and $D_{\text{trans}}$ and populates them purely via a counterexample-guided procedure, which identifies edge-cases one by one towards the synthesis of neural partially-ranking functions. This is in contrast with the cognate neural model checking approach based on gradient descent, which benefits from random test generation to initially populate the datasets and rarely relies on the generation of data by means of counterexamples [63, Section 5]. Figure 4b evaluates our MILP-based approach under multiple random initialisation sizes with 0 (our default), 10, 25, and 50 randomly generated samples, respectively. As our experiment indicates, our default approach is nearly indistinguishable from the *Best* curve (which indicates the fastest configuration per instance). This indicates that our MILP-based approach does not benefit from unguided random data generation.

**Comparison with SMT Learning Engines**   Our new architecture makes learning amenable to MILP solving and, more generally, to decision procedures for linear theories implemented in SMT solvers. Figure 4c compares alternative engines for our learn phase, where we replace the MILP solver `Gurobi` with the SMT solvers `CVC5` 1.1.2 [14], `MathSAT` 5.6.10 [36], and `Z3` 4.13.0 [49] via `PySMT` [59]. Within the $5\,\text{h}$ timeout, `MathSAT` completed $81\,\%$ of tasks, `Gurobi` completed $87\,\%$, `CVC5` completed $88\,\%$, and `Z3` completed $96\,\%$ of tasks. `Gurobi` and `Z3` offer the best compromise between speed and efficacy, with `Z3` demonstrating superior coverage and `Gurobi` shorter runtimes: `Gurobi` was on average $7\times$ faster than `Z3` on successfully solved instances. However, `Gurobi` failed to return a result in $13\,\%$ of cases because of memory segmentation faults, model infeasibility for our hyper-parameters, numerical mismatches after evaluation of the result with `NumPy`; only 2 tasks were timeouts. The *Best* curve selects the fastest engine per instance—obviously outperforming every other solver, providing a baseline for a potential portfolio implementation. We remark that all experiments besides this comparison (Figure 4c) use only `Gurobi` as its learning engine.

**Strengths and Limitations**   Our approach is much more straightforward in terms of parameter tuning then the prior neural model checking approach [63], owing to the theoretical completeness and the superior numerical stability of MILP and SMT algorithms over stochastic gradient descent. However, this strongly relies on sign-activated feed-forward neural networks; the generalisation to further architectures is an open problem for future investigation. Our method is amenable to training over parameters of integer type, which naturally results in quantised neural partially-raking functions. On the other hand, our approach is limited to hardware and reactive systems without arrays or strings of parametric length, procedure calls or dynamic data structures, which are topics for future work.

**Threats to Validity**   Our empirical evaluation has demonstrated the superior performance of our approach over the state of the art on verification tasks derived from standard textbook level hardware designs. Our benchmark suite encompasses word-level designs with safety and liveness properties, unlike the standard HWMCC'24 benchmarks which focus on pure safety properties [19]. Our benchmark is arguably the hardest (as of today) for word-level liveness and combined safety-liveness hardware verification. Yet, our evaluation may not be representative of verifications tasks other than our suite and further research is required to assess the generalisability and the scalability of our approach to other workloads. Future work includes the integration of neural and compositional model checking towards the scalable formal verification of industrial-scale designs [40, 88, 134].

# 6 Related Work

**Model Checking Linear Temporal Logic**    Formal hardware verification relies on model checking LTL specifications—codified, e.g., as SystemVerilog Assertions—to guarantee safety and liveness properties [8, 56, 77, 102]. Symbolic engines based on BDD fixed-points [10, 41, 75] or SAT/IC3 reasoning [17, 20, 73, 87, 114] among other methods have evolved over fifty years, with key contributions to formal verification honoured by the ACM Turing Awards of 1996, 2007, and 2013. The automata-theoretic approaches to model checking often reduce model checking to fair-emptiness. This has been tackled via $k$-liveness bounded model checking [38, 69], IC3 with strongly connected components [23], or BDDs with Emerson–Lei fixed-point computation [52]. Ranking functions were originally devised for program termination [55] and later generalised to liveness certificates [5, 6, 44, 50, 64, 81, 125], may be linear [21, 104], piecewise-defined [76, 123, 124], word-level [32, 47], lexicographic [22, 82], or represented as disjunctive well-founded relations [37, 43, 45, 74, 103]. Traditional algorithms for the automated construction of ranking functions and inductive invariants usually rely on constraint solving (using Farkas' lemma or Positivstellensatz results) or abstract interpretation [31, 43, 45, 74, 103, 105].

**Machine Learning for Automated Reasoning**    Neural methods have permeated every layer of formal reasoning. In theorem proving, they guide clause selection [53, 84], premise selection [13, 68, 89, 127], tactic prediction [100, 132], and even end-to-end proof search [54, 106]. In constraint solving, neural approaches have been integrated into SMT [12, 85, 112] and SAT solving [61, 79, 113, 128], and have been further extended to combinatorial optimisation [83] and MILP [60, 93]. Similar ideas have been applied to program synthesis [15, 33, 72, 122], algorithm selection [80, 86], termination analysis [7], circuit synthesis and repair from temporal logic specifications [46, 111], and generation of satisfying traces for given LTL properties [66]. A large body of work focussed on learning inductive invariants via teacher–learner loops [57], decision trees [58, 76], random search [115], data-driven templates [98], neural networks [109, 133], reinforcement learning [116, 131] and, most recently, large language models [29, 71, 101, 130]. These techniques target safety alone; LTL specifications that encompass both safety *and* liveness are out of scope. Handling properties that combine safety and liveness using machine learning fills a significant gap in the field.

**Neural Certificates**    As opposed to using neural networks to *generate* formal proof certificates, our approach uses neural networks to *represent* them (along with mathematical guarantees), falling within the spectrum of neural certificates. This extends recent work on control [2, 30, 48, 78, 92, 95, 108, 135–137], formal verification of software and probabilistic programs [3, 4, 62], and model checking reactive systems under *pure liveness* LTL properties [63]. Our work has enabled neural certificates to effectively prove *arbitrary* LTL properties, orders of magnitude faster than the state of the art.

# 7 Conclusion

We have presented a novel and improved neural model checking approach for arbitrary compositions of safety and liveness specifications. By introducing a specialised neural partially-ranking function architecture that (i) simultaneously represents inductive invariants and ranking functions and (ii) is amenable to training with numerically stable MILP solvers in a counterexample-guided procedure, our method yields compact and naturally quantised certificates. We implemented a lightweight Python prototype that combines off-the-shelf MILP (`Gurobi`), SMT (`Bitwuzla`), and BMC (`EBMC`) to train and certify the network in an iterative approach that provides formal guarantees [65, 91, 96]. Our improved technique attains speedups up to $10^5 \times$ and $10\,\%$ higher completion (within a $5\,\mathrm{h}$ time limit) over the state-of-the-art symbolic model checkers and prior work on neural model checking [24, 28, 63, 119], on a benchmark suite of 634 hardware model checking tasks encompassing safety, liveness and combined safety-liveness verification tasks written in SystemVerilog.

A complete specification for the functional correctness of reactive systems must capture *both* safety and liveness: the former guarantees the absence of undesired behaviour; the latter guarantees the presence of desired behaviour. Our approach has extended neural model checking to capture both safety and liveness, simultaneously. This paves a path towards the provably safe application of artificial intelligence and machine learning to the design of high-assurance industrial hardware, with impact to avionics [90, 118], automotive [97, 120], medical devices [70, 126], nuclear control [27, 99], rail signalling [34, 35], robotics [110, 129], utility grids [25, 94], and more.

## Acknowledgments

We thank Matthew Leeke, Sonia Marin, and Mark Ryan for their feedback and the anonymous reviewers for their comments and suggestions on this manuscript. This work was supported in part by the Amazon Research Award *Neural Software Verification* (Fall 2024), and by the Advanced Research and Invention Agency (ARIA) under the *Safeguarded AI* programme.

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

| Model | LTL Specification (Pure-Liveness) |
|---|---|
| DELAY | $\mathsf{FG}\ \texttt{!rst} \to \mathsf{GF}\ \texttt{sig}$ |
| | $\mathsf{FG}\ \texttt{!rst} \to \mathsf{GF}\ \left(\texttt{sig} \land \mathsf{X}\ \texttt{!sig}\right)$ |
| LCD Controller | $\mathsf{FG}\ \texttt{lcd\_enable} \to \mathsf{GF}\ \texttt{ready}$ |
| Blink | $\mathsf{FG}\ \texttt{!rst} \to \mathsf{GF}\ \texttt{ledON}$ |
| Thermocouple | $\mathsf{FG}\ \texttt{!rst} \to \mathsf{GF}\ \texttt{get\_data}$ |
| 7-Segment | $\mathsf{FG}\ \texttt{!rst} \to \mathsf{GF}\ \texttt{disp = 1}$ |
| | $\mathsf{FG}\ \texttt{!rst} \to \left(\mathsf{GF}\ \texttt{disp = 0} \land \mathsf{GF}\ \texttt{disp = 1}\right)$ |
| i2c Stretch | $\mathsf{FG}\ \left(\texttt{!rst} \land \texttt{ena}\right) \to \mathsf{GF}\ \texttt{strc}$ |
| Pulse Width Modulation | $\mathsf{GF}\ \texttt{!pulse}$ |
| VGA Controller | $\mathsf{FG}\ \texttt{!rst} \to \mathsf{GF}\ \texttt{disp\_ena}$ |
| UART Transmitter | $\mathsf{FG}\ \texttt{!rst} \to \mathsf{GF}\ \texttt{wait}$ |
| Load-Store | $\mathsf{FG}\ \texttt{!rst} \to \mathsf{GF}\ \texttt{sig}$ |
| Gray Counter | $\mathsf{FG}\ \texttt{!rst} \to \mathsf{GF}\ \texttt{sig}$ |
| | $\mathsf{FG}\ \texttt{!rst} \to \mathsf{GF}\ \left(\texttt{sig} \land \mathsf{X}\ \texttt{!sig}\right)$ |
| | $\mathsf{FG}\ \texttt{!rst} \to \left(\mathsf{GF}\ \texttt{sig} \land \mathsf{GF}\ \texttt{!sig}\right)$ |

Table 2: Model Name and LTL Specification in our Benchmark (Pure-Liveness)

| Model | LTL Specification (Pure-Safety) |
|---|---|
| DELAY | $\mathsf{G}\ \texttt{!err}$ |
| | $\mathsf{G}\ \left(\texttt{sig} \to \mathsf{X}\ \texttt{!sig}\right)$ |
| LCD Controller | $\mathsf{G}\ \left(\left(\texttt{!lcd\_enable} \land \texttt{ready}\right) \to \mathsf{X}\ \texttt{ready}\right)$ |
| | $\mathsf{G}\ \left(\mathsf{X}\ \texttt{wait} \to \texttt{busy}\right)$ |
| Blink | $\mathsf{G}\ \left(\left(\texttt{mode1} \land \mathsf{X}\ \texttt{mode1}\right) \to \mathsf{X}\ \left(\texttt{cnt!=0}\right)\right)$ |
| Thermocouple | $\mathsf{G}\ \left(\left(\texttt{!spi\_busy} \land \texttt{get\_data} \land \texttt{!rst}\right) \to \mathsf{X}\ \texttt{get\_data}\right)$ |
| 7-Segment | $\mathsf{G}\ \left(\left(\texttt{!sig} \land \mathsf{X}\ \texttt{!sig} \land \texttt{!rst}\right) \to \left(\left(\texttt{ds0} \land \mathsf{X}\ \texttt{ds0}\right) \lor \left(\texttt{ds1} \land \mathsf{X}\ \texttt{ds1}\right)\right)\right)$ |
| i2c Stretch | $\mathsf{G}\ \left(\left(\texttt{!strc} \land \mathsf{X}\ \texttt{strc}\right) \to \mathsf{X}\ \texttt{switch}\right)$ |
| Pulse Width Modulation | $\mathsf{G}\ \left(\left(\texttt{pulseLB} \to \texttt{pulse}\right) \land \left(\texttt{!pulseUB} \to \texttt{!pulse}\right)\right)$ |
| VGA Controller | $\mathsf{G}\ \left(\left(\texttt{en} \land \texttt{!rst}\right) \to \left(\left(\texttt{hs} \leftrightarrow \mathsf{X}\ \texttt{!hs}\right) \lor \left(\texttt{!hs} \leftrightarrow \mathsf{X}\ \texttt{hs}\right)\right)\right)$ |
| | $\mathsf{G}\ \left(\left(\texttt{en} \land \texttt{!rst}\right) \to \left(\left(\texttt{vs} \leftrightarrow \mathsf{X}\ \texttt{!vs}\right) \lor \left(\texttt{!vs} \leftrightarrow \mathsf{X}\ \texttt{vs}\right)\right)\right)$ |
| UART Transmitter | $\mathsf{G}\ \left(\left(\texttt{wait} \to \texttt{!busy}\right) \land \left(\texttt{transmit} \to \texttt{busy}\right)\right)$ |
| Load-Store | $\mathsf{G}\ \left(\texttt{sig} \to \mathsf{X}\ \mathsf{X}\ \texttt{!sig}\right)$ |
| Gray Counter | $\mathsf{G}\ \left(\left(\texttt{sig} \land \texttt{!rst}\right) \to \mathsf{X}\ \texttt{!sig}\right)$ |

Table 3: Model Name and LTL Specification in our Benchmark (Pure-Safety)

| Model | LTL Specification (Safety-Liveness) |
|---|---|
| Delay | $\mathsf{G}\,!\mathtt{rst} \to \mathsf{XG}\left(cnt < N\,\mathsf{U}\,\mathtt{sig}\right)$ |
| | $\mathsf{FG}\,!\mathtt{rst} \to \mathsf{FG}\left(cnt < N\,\mathsf{U}\,\mathtt{sig}\right)$ |
| LCD Controller | $\mathsf{G}\left(\mathtt{receive} \to \left(\mathtt{receive}\,\mathsf{U}\,\mathtt{ready}\right)\right)$ |
| Blink | $\mathsf{G}\,!\mathtt{rst} \to \mathsf{G}\left(\mathtt{ledON} \to \left(\mathtt{ledON}\,\mathsf{U}\,\mathtt{mode0}\right)\right)$ |
| Thermocouple | $\mathsf{G}\,!\mathtt{rst} \to \mathsf{G}\left(\mathtt{pause} \to \left(\mathtt{pause}\,\mathsf{U}\,\mathtt{get\_data}\right)\right)$ |
| 7-Segment | $\mathsf{G}\,!\mathtt{rst} \to \mathsf{G}\left(\left(\left(\mathtt{ds0} \wedge \mathsf{X}\,\mathtt{ds0}\right) \vee \left(\mathtt{ds1} \wedge \mathsf{X}\,\mathtt{ds1}\right)\right)\mathsf{U}\,\mathtt{sig}\right)$ |
| i2c Stretch | $\mathsf{G}\,!\mathtt{rst} \to \mathsf{G}\left(!\mathtt{strc} \to \left(!\mathtt{strc}\,\mathsf{U}\,\mathtt{switch}\right)\right)$ |
| Pulse Width Modulation | $\mathsf{GF}\,!\mathtt{pulseUB} \wedge \mathsf{XG}\left(!\mathtt{pulseUB} \to\, !\mathtt{pulse}\right)$ |
| VGA Controller | $\mathsf{G}\left(\left(\mathtt{Vcnt == 0}\right) \to \left(\left(\mathtt{Vcnt == 0}\right)\mathsf{U}\left(\mathtt{Hcnt == 0}\right)\right)\right)$ |
| UART Transmitter | $\mathsf{G}\,!\mathtt{rst} \to \mathsf{G}\left(\mathtt{busy} \to \left(\mathtt{busy}\,\mathsf{U}\,\mathtt{wait}\right)\right)$ |
| Load-Store | $\mathsf{G}\,!\mathtt{rst} \to \mathsf{G}\left(\mathtt{modeUP} \to \left(\mathtt{modeUP}\,\mathsf{U}\,\mathtt{sig}\right)\right)$ |
| | $\mathsf{G}\,!\mathtt{rst} \to \mathsf{X}\,\mathsf{G}\left(\mathtt{modeUP} \to \left(!\mathtt{sig}\,\mathsf{U}\,\mathtt{modeDOWN}\right)\right)$ |
| Gray Counter | $\mathsf{G}\,!\mathtt{rst} \to \mathsf{XG}\left(cnt > 0\,\mathsf{U}\,\mathtt{sig}\right)$ |
| | $\mathsf{FG}\,!\mathtt{rst} \to \mathsf{FG}\left(cnt > 0\,\mathsf{U}\,\mathtt{sig}\right)$ |

Table 4: Model Name and LTL Specification in our Benchmark (Safety + Liveness)

## A  Details of the Case Studies

We benchmark our tool on eleven RTL designs. Ten are adopted from the neural model-checking study of [63], which provided *only* pure-liveness properties. We extend every design with semantically natural pure safety and safety plus liveness specifications. Table 2 lists all pure-liveness formulas, Table 3 the pure-safety formulas, and Table 4 the safety-liveness formulas, all in Linear Temporal Logic (LTL). Below we discuss each design together with its full specification suite.

**Delay** (DELAY). A counter cnt raises a pulse sig after a fixed delay and is reset by rst. Liveness requires $\mathsf{FG}\,!\mathtt{rst} \to \mathsf{GF}\,\mathtt{sig}$ and, to prevent sig from sticking, $\mathsf{FG}\,!\mathtt{rst} \to \mathsf{GF}\,(\mathtt{sig} \wedge \mathsf{X}\,!\mathtt{sig})$. Safety bounds the counter, $\mathsf{G}\,cnt <= N$, and clears sig in the next cycle, $\mathsf{G}\,(\mathtt{sig} \to \mathsf{X}\,!\mathtt{sig})$. Safety-Liveness formulas constrain cnt until sig occurs, $\mathsf{G}\,!\mathtt{rst} \to \mathsf{XG}\,(cnt < N\,\mathsf{U}\,\mathtt{sig})$ and its eventually variant $\mathsf{FG}\,!\mathtt{rst} \to \mathsf{FG}\,(cnt < N\,\mathsf{U}\,\mathtt{sig})$.

**LCD Controller** (LCD). After initialisation the controller waits for lcd_enable, switches from ready to send, and returns to ready. Liveness demands $\mathsf{FG}\,\mathtt{lcd\_enable} \to \mathsf{GF}\,\mathtt{ready}$. Safety preserves ready when lcd_enable is low, $\mathsf{G}\big((!\mathtt{lcd\_enable} \wedge \mathtt{ready}) \to \mathsf{X}\,\mathtt{ready}\big)$, and asserts busy in wait, $\mathsf{G}\big(\mathsf{X}\,\mathtt{wait} \to\ \mathtt{busy}\big)$; for safety-liveness a receive phase persists until ready, $\mathsf{G}\big(\mathtt{receive} \to (\mathtt{receive}\,\mathsf{U}\,\mathtt{ready})\big)$.

**Blink** (BLINK). A counter toggles led on every wrap and raises a one-cycle flag flg. Liveness asserts $\mathsf{FG}\,!\mathtt{rst} \to \mathsf{GF}\,\mathtt{ledON}$, while safety enforces staying in mode1 if cnt has not wrapped: $\mathsf{G}\big((\mathtt{mode1} \wedge \mathsf{X}\,\mathtt{mode1}) \to \mathsf{X}(cnt \neq 0)\big)$. The safety–liveness specification ensures that led remains on until mode0 is reached: $\mathsf{G}\,!\mathtt{rst} \to \mathsf{G}\big(\mathtt{ledON} \to (\mathtt{ledON}\,\mathsf{U}\,\mathtt{mode0})\big)$.

**Thermocouple** (Tmcp.). The FSM cycles through start, get_data, pause. Liveness requires $\mathsf{FG}\,!\mathtt{rst} \to \mathsf{GF}\,\mathtt{get\_data}$. Safety keeps get_data active on an idle bus and safety-liveness keeps pause active until get_data becomes reachable: $\mathsf{G}\big(!\mathtt{spi\_busy} \wedge \mathtt{get\_data} \wedge\, !\mathtt{rst} \to \mathsf{X}\,\mathtt{get\_data}\big)$ and $\mathsf{G}\,!\mathtt{rst} \to \mathsf{G}\big(\mathtt{pause} \to (\mathtt{pause}\,\mathsf{U}\,\mathtt{get\_data})\big)$.

**7-Segment** (7-Seg). Two displays alternate unless reset. Liveness enforces $\mathsf{FG}$ !rst $\rightarrow$ ($\mathsf{GF}$ disp = 0 $\wedge$ $\mathsf{GF}$ disp = 1) and the single-display variant $\mathsf{FG}$ !rst $\rightarrow$ $\mathsf{GF}$ disp = 1. Safety freezes the display in the absence of sig: $\mathsf{G}\big($!sig$\wedge\mathsf{X}$!sig$\wedge$!rst $\rightarrow$ ((ds0$\wedge\mathsf{X}$ds0)$\vee$(ds1$\wedge\mathsf{X}$ds1))$\big)$ while safety-liveness holds the display until sig, $\mathsf{G}$ !rst $\rightarrow$ $\mathsf{G}\big(((\text{ds0} \wedge \mathsf{X}\text{ds0}) \vee (\text{ds1} \wedge \mathsf{X}\text{ds1})) \mathsf{U} \text{ sig}\big)$.

**i²c Stretch** (i2cS). Timing signals scl_clk and data_clk are generated according to bus frequency; stretch handles clock stretching. Liveness states $\mathsf{FG}$ (!rst $\wedge$ ena) $\rightarrow$ $\mathsf{GF}$ strc. Safety links a rising stretch to switch: $\mathsf{G}\big($!strc $\wedge$ $\mathsf{X}$strc $\rightarrow$ $\mathsf{X}$switch$\big)$ meanwhile safety-liveness, under permanent reset-low, keeps stretch low until switch, $\mathsf{G}$ !rst $\rightarrow$ $\mathsf{G}\big($!strc $\rightarrow$ (!strc $\mathsf{U}$ switch)$\big)$.

**Pulse Width Modulation** (PWM). An $N$-bit counter drives pulse; the design must regularly unset the pulse. Accordingly, liveness requires $\mathsf{GF}$ !pulse. Two guard signals bound the duty-cycle window: pulseLB (minimum width) and pulseUB (maximum width). Safety enforces these bounds at every cycle, $\mathsf{G}\big(($pulseLB$\rightarrow$pulse$)$ $\wedge$ $($!pulseUB$\rightarrow$!pulse$)\big)$, and safety-liveness strengthens them with a fairness clause that forces the upper bound to be violated—and thus pulse to be low—infinitely often: $\mathsf{GF}$ !pulseUB $\wedge$ $\mathsf{XG}\big($!pulseUB$\rightarrow$!pulse$\big)$.

**VGA Controller** (VGA). Horizontal and vertical counters drive disp_ena. Liveness: $\mathsf{FG}$ !rst $\rightarrow$ $\mathsf{GF}$ disp_ena. Safety flips hs and vs under disp_ena: $\mathsf{G}\big(($en $\wedge$ !rst$)$ $\rightarrow$ $(($hs $\leftrightarrow$ $\mathsf{X}$!hs$)$ $\vee$ $($!hs $\leftrightarrow$ $\mathsf{X}$hs$))\big)$, $\mathsf{G}\big(($en $\wedge$ !rst$)$ $\rightarrow$ $(($vs $\leftrightarrow$ $\mathsf{X}$!vs$)$ $\vee$ $($!vs $\leftrightarrow$ $\mathsf{X}$vs$))\big)$, and safety-liveness asserts a vertical–horizontal counter relationship $\mathsf{G}\big($Vcnt==0 $\rightarrow$ $($Vcnt==0 $\mathsf{U}$ Hcnt==0$)\big)$.

**UART Transmitter** (UARTt). The FSM toggles between wait and transmit. Liveness: $\mathsf{FG}$ !rst $\rightarrow$ $\mathsf{GF}$ wait. Safety couples busy with required modes: $\mathsf{G}\big(($wait $\rightarrow$ !busy$)$ $\wedge$ $($transmit $\rightarrow$ busy$)\big)$, and safety-liveness under permanent reset-low holds busy until wait, $\mathsf{G}$ !rst $\rightarrow$ $\mathsf{G}\big($busy $\rightarrow$ $($busy $\mathsf{U}$ wait$)\big)$.

**Load–Store** (LS). A counter counts up in load, down in store; sig triggers the switch. Liveness: $\mathsf{FG}$ !rst $\rightarrow$ $\mathsf{GF}$ sig. Safety clears sig after two cycles: $\mathsf{G}\big($sig $\rightarrow$ $\mathsf{XX}$!sig$\big)$. The safety-liveness formulas hold modeUP until sig and, symmetrically, !sig until modeDOWN: $\mathsf{G}$ !rst $\rightarrow$ $\mathsf{G}\big($modeUP $\rightarrow$ $($modeUP $\mathsf{U}$ sig$)\big)$ and $\mathsf{G}$ !rst $\rightarrow$ $\mathsf{XG}\big($modeUP $\rightarrow$ $($!sig $\mathsf{U}$ modeDOWN$)\big)$.

**Gray Counter** (Gray). Counts in Gray code to minimise bit flips. Liveness: $\mathsf{FG}$ !rst $\rightarrow$ $\mathsf{GF}$ sig, $\mathsf{FG}$!rst $\rightarrow$ $\mathsf{GF}($sig $\wedge$ $\mathsf{X}$!sig$)$, and $\mathsf{FG}$!rst $\rightarrow$ $(\mathsf{GF}$sig $\wedge$ $\mathsf{GF}$!sig$)$. Safety clears sig one step later: $\mathsf{G}\big($sig $\wedge$ !rst $\rightarrow$ $\mathsf{X}$!sig$)$. Safety-liveness bind the counter until sig: $\mathsf{G}$ !rst $\rightarrow$ $\mathsf{XG}(cnt > 0$ $\mathsf{U}$ sig$)$ and its eventually counterpart $\mathsf{FG}$ !rst $\rightarrow$ $\mathsf{FG}(cnt > 0$ $\mathsf{U}$ sig$)$.

Collectively, these benchmarks exercise handshake logic, clock stretching, display timing, PWM generation, UART serialisation, counter bounds, and Gray-code sequencing. Each property appears in pure-liveness, pure-safety, and safety-liveness form, enabling a comprehensive assessment across the complete verification spectrum.

# NeurIPS Paper Checklist

1. **Claims**

   Question: Do the main claims made in the abstract and introduction accurately reflect the paper's contributions and scope?

   Answer: [Yes]

   Justification: The main claims of this paper are stated in the abstract and elaborated in Section 1. There, we briefly outline the theoretical underpinnings of these claims and summarise experimental results that quantify the scalability of our approach.

   Guidelines:

   - The answer NA means that the abstract and introduction do not include the claims made in the paper.
   - The abstract and/or introduction should clearly state the claims made, including the contributions made in the paper and important assumptions and limitations. A No or NA answer to this question will not be perceived well by the reviewers.
   - The claims made should match theoretical and experimental results, and reflect how much the results can be expected to generalise to other settings.
   - It is fine to include aspirational goals as motivation as long as it is clear that these goals are not attained by the paper.

2. **Limitations**

   Question: Does the paper discuss the limitations of the work performed by the authors?

   Answer: [Yes]

   Justification: Section 5 includes a dedicated *Strengths and Limitations* subsection, where we explicitly state our drawbacks and justify their reasonableness, along with a discussion of theoretical and practical constraints. Section 5 contains a class of problems (pure-safety) where we underperform. The accompanying *Threats to Validity* subsection critically evaluates the scope of our claims by motivating the benchmark selection used for comparison against alternative verification methods.

   Guidelines:

   - The answer NA means that the paper has no limitation while the answer No means that the paper has limitations, but those are not discussed in the paper.
   - The authors are encouraged to create a separate "Limitations" section in their paper.
   - The paper should point out any strong assumptions and how robust the results are to violations of these assumptions (e.g., independence assumptions, noiseless settings, model well-specification, asymptotic approximations only holding locally). The authors should reflect on how these assumptions might be violated in practice and what the implications would be.
   - The authors should reflect on the scope of the claims made, e.g., if the approach was only tested on a few datasets or with a few runs. In general, empirical results often depend on implicit assumptions, which should be articulated.
   - The authors should reflect on the factors that influence the performance of the approach. For example, a facial recognition algorithm may perform poorly when image resolution is low or images are taken in low lighting. Or a speech-to-text system might not be used reliably to provide closed captions for online lectures because it fails to handle technical jargon.
   - The authors should discuss the computational efficiency of the proposed algorithms and how they scale with dataset size.
   - If applicable, the authors should discuss possible limitations of their approach to address problems of privacy and fairness.
   - While the authors might fear that complete honesty about limitations might be used by reviewers as grounds for rejection, a worse outcome might be that reviewers discover limitations that aren't acknowledged in the paper. The authors should use their best judgment and recognize that individual actions in favor of transparency play an important role in developing norms that preserve the integrity of the community. Reviewers will be specifically instructed to not penalize honesty concerning limitations.

3. **Theory assumptions and proofs**

Question: For each theoretical result, does the paper provide the full set of assumptions and a complete (and correct) proof?

Answer: [Yes]

Justification: We provide references for all theoretical foundations underpinning our work, including automata-theoretic LTL model checking, fair termination, inductive invariant and the use of ranking functions and inductive invariants in sec 2.

Guidelines:

- The answer NA means that the paper does not include theoretical results.
- All the theorems, formulas, and proofs in the paper should be numbered and cross-referenced.
- All assumptions should be clearly stated or referenced in the statement of any theorems.
- The proofs can either appear in the main paper or the supplemental material, but if they appear in the supplemental material, the authors are encouraged to provide a short proof sketch to provide intuition.
- Inversely, any informal proof provided in the core of the paper should be complemented by formal proofs provided in appendix or supplemental material.
- Theorems and Lemmas that the proof relies upon should be properly referenced.

4. **Experimental result reproducibility**

Question: Does the paper fully disclose all the information needed to reproduce the main experimental results of the paper to the extent that it affects the main claims and/or conclusions of the paper (regardless of whether the code and data are provided or not)?

Answer: [Yes]

Justification: The *Implementation* subsection of Section 5 describes the full pipeline used in our prototype and specifies the versions of all external dependencies. It outlines the neural network architecture under both linear and automatic configurations, as well as the learning engines used. This section also states the configurations employed in our main experiments and ablation studies. In contrast, Section 3 addresses the theoretical underpinnings of the learning procedure, including dataset generation, the design of the neural architecture, and the SMT-based check step, as part of our iterative approach. Together, these sections provide sufficient detail to replicate our method for neural model checking of safety and liveness properties. Additionally, the *Experimental Setup* subsection discusses the versions of competing tools, the hardware environment used for evaluation, and the translation procedure from SystemVerilog to the appropriate input formats for each tool. For reproducibility and further implementation details, our code and benchmarks will be made publicly available.

Guidelines:

- The answer NA means that the paper does not include experiments.
- If the paper includes experiments, a No answer to this question will not be perceived well by the reviewers: Making the paper reproducible is important, regardless of whether the code and data are provided or not.
- If the contribution is a dataset and/or model, the authors should describe the steps taken to make their results reproducible or verifiable.
- Depending on the contribution, reproducibility can be accomplished in various ways. For example, if the contribution is a novel architecture, describing the architecture fully might suffice, or if the contribution is a specific model and empirical evaluation, it may be necessary to either make it possible for others to replicate the model with the same dataset, or provide access to the model. In general. releasing code and data is often one good way to accomplish this, but reproducibility can also be provided via detailed instructions for how to replicate the results, access to a hosted model (e.g., in the case of a large language model), releasing of a model checkpoint, or other means that are appropriate to the research performed.
- While NeurIPS does not require releasing code, the conference does require all submissions to provide some reasonable avenue for reproducibility, which may depend on the nature of the contribution. For example

(a) If the contribution is primarily a new algorithm, the paper should make it clear how to reproduce that algorithm.

(b) If the contribution is primarily a new model architecture, the paper should describe the architecture clearly and fully.

(c) If the contribution is a new model (e.g., a large language model), then there should either be a way to access this model for reproducing the results or a way to reproduce the model (e.g., with an open-source dataset or instructions for how to construct the dataset).

(d) We recognize that reproducibility may be tricky in some cases, in which case authors are welcome to describe the particular way they provide for reproducibility. In the case of closed-source models, it may be that access to the model is limited in some way (e.g., to registered users), but it should be possible for other researchers to have some path to reproducing or verifying the results.

5. **Open access to data and code**

Question: Does the paper provide open access to the data and code, with sufficient instructions to faithfully reproduce the main experimental results, as described in supplemental material?

Answer: [Yes]

Justification: Alongside the paper, we provide a zip archive as per NeurIPS guidelines. It includes all benchmarks, experiment scripts, and a `README.md` with detailed instructions for reproducing our results.

Guidelines:

- The answer NA means that paper does not include experiments requiring code.
- Please see the NeurIPS code and data submission guidelines (`https://nips.cc/public/guides/CodeSubmissionPolicy`) for more details.
- While we encourage the release of code and data, we understand that this might not be possible, so "No" is an acceptable answer. Papers cannot be rejected simply for not including code, unless this is central to the contribution (e.g., for a new open-source benchmark).
- The instructions should contain the exact command and environment needed to run to reproduce the results. See the NeurIPS code and data submission guidelines (`https://nips.cc/public/guides/CodeSubmissionPolicy`) for more details.
- The authors should provide instructions on data access and preparation, including how to access the raw data, preprocessed data, intermediate data, and generated data, etc.
- The authors should provide scripts to reproduce all experimental results for the new proposed method and baselines. If only a subset of experiments are reproducible, they should state which ones are omitted from the script and why.
- At submission time, to preserve anonymity, the authors should release anonymized versions (if applicable).
- Providing as much information as possible in supplemental material (appended to the paper) is recommended, but including URLs to data and code is permitted.

6. **Experimental setting/details**

Question: Does the paper specify all the training and test details (e.g., data splits, hyperparameters, how they were chosen, type of optimizer, etc.) necessary to understand the results?

Answer: [Yes]

Justification: Section 5, particularly the *Implementation* subsection, discusses our use of a single hyperparameter and outlines how it is selected. We also detail the tools used for the *learn* and *check* phases. Our approach to dataset generation is described in Section 3, and a full description of the benchmark suite is provided in Appendix A.

Guidelines:

- The answer NA means that the paper does not include experiments.
- The experimental setting should be presented in the core of the paper to a level of detail that is necessary to appreciate the results and make sense of them.

- The full details can be provided either with the code, in appendix, or as supplemental material.

7. **Experiment statistical significance**

Question: Does the paper report error bars suitably and correctly defined or other appropriate information about the statistical significance of the experiments?

Answer: [Yes]

Justification: Because formal-verification tools are deterministic and must return provably correct answers, traditional variance measures—error bars and confidence intervals—are not meaningful. Instead for statistical significance, we run each tool once on a substantial benchmark suite of 634 instances and provide cactus plots, an experimental setup consistent with community practice in formal verification competitions such as HWMCC. Statistical evidence is conveyed by cactus plots (Figure 3(a–c); Figure 4(b-c)), which report, for every time cap up to $15\,\mathrm{min}$, the cumulative number of tasks each competitor completes. Scatter plots (Figure 3(d–f)) complement this view by plotting one point per benchmark task, exposing the full runtime distribution rather than aggregate summaries. We additionally annotate salient figures in our text—e.g., the percentage of tasks finished within a given cap and the count lying to the right of the $10\times$ diagonal—to underscore the consistent performance gaps observed across tools and throughout our ablation study.

Guidelines:

- The answer NA means that the paper does not include experiments.
- The authors should answer "Yes" if the results are accompanied by error bars, confidence intervals, or statistical significance tests, at least for the experiments that support the main claims of the paper.
- The factors of variability that the error bars are capturing should be clearly stated (for example, train/test split, initialization, random drawing of some parameter, or overall run with given experimental conditions).
- The method for calculating the error bars should be explained (closed form formula, call to a library function, bootstrap, etc.)
- The assumptions made should be given (e.g., Normally distributed errors).
- It should be clear whether the error bar is the standard deviation or the standard error of the mean.
- It is OK to report 1-sigma error bars, but one should state it. The authors should preferably report a 2-sigma error bar than state that they have a 96% CI, if the hypothesis of Normality of errors is not verified.
- For asymmetric distributions, the authors should be careful not to show in tables or figures symmetric error bars that would yield results that are out of range (e.g. negative error rates).
- If error bars are reported in tables or plots, The authors should explain in the text how they were calculated and reference the corresponding figures or tables in the text.

8. **Experiments compute resources**

Question: For each experiment, does the paper provide sufficient information on the computer resources (type of compute workers, memory, time of execution) needed to reproduce the experiments?

Answer: [Yes]

Justification: The compute environment is described in the *Experimental Setup* subsection of Section 5, where we specify our Amazon EC2 instance (type and configuration) and report the total wall-clock time across all experiments. To avoid cluttering the paper with over 7 000 individual measurements (634 tasks × multiple settings), we provide the complete runtime data in the supplementary ZIP archive and through cactus plots. In the main paper, one can visualise the per-task runtime through the cactus (Figure 3(a–c);Figure 4(b-c)) and scatter plots 3(d-f)).

Guidelines:

- The answer NA means that the paper does not include experiments.

- The paper should indicate the type of compute workers CPU or GPU, internal cluster, or cloud provider, including relevant memory and storage.
- The paper should provide the amount of compute required for each of the individual experimental runs as well as estimate the total compute.
- The paper should disclose whether the full research project required more compute than the experiments reported in the paper (e.g., preliminary or failed experiments that didn't make it into the paper).

9. **Code of ethics**

Question: Does the research conducted in the paper conform, in every respect, with the NeurIPS Code of Ethics https://neurips.cc/public/EthicsGuidelines?

Answer: [Yes]

Justification: We affirm that all content presented in this work is disclosed with full academic integrity. All original contributions are properly cited, and our methodology is described in detail throughout the paper to ensure reproducibility. In addition, we provide our full implementation and experimental setup as part of the supplementary materials. Our experiments exclusively use synthetic data, with no human subjects involved, and are therefore fully aligned with ethical research standards. The goal of our method is to enhance the reliability and safety of computer systems. The benchmark hardware designs are drawn from prior work, which is cited accordingly, with one additional design introduced by us. While the pure-liveness specifications are taken from the same work, we developed additional pure-safety and safety-liveness specifications based on the semantics of each design. We have reviewed the NeurIPS Code of Ethics and confirm our strict adherence to its principles.

Guidelines:

- The answer NA means that the authors have not reviewed the NeurIPS Code of Ethics.
- If the authors answer No, they should explain the special circumstances that require a deviation from the Code of Ethics.
- The authors should make sure to preserve anonymity (e.g., if there is a special consideration due to laws or regulations in their jurisdiction).

10. **Broader impacts**

Question: Does the paper discuss both potential positive societal impacts and negative societal impacts of the work performed?

Answer: [Yes]

Justification: As discussed in the introduction and conclusion, improving the correctness of hardware designs prior to fabrication contributes to the development of safer and more reliable systems. Avoiding the manufacture of flawed silicon also reduces material waste. We are not aware of any direct path by which our method could be applied to harmful ends.

Guidelines:

- The answer NA means that there is no societal impact of the work performed.
- If the authors answer NA or No, they should explain why their work has no societal impact or why the paper does not address societal impact.
- Examples of negative societal impacts include potential malicious or unintended uses (e.g., disinformation, generating fake profiles, surveillance), fairness considerations (e.g., deployment of technologies that could make decisions that unfairly impact specific groups), privacy considerations, and security considerations.
- The conference expects that many papers will be foundational research and not tied to particular applications, let alone deployments. However, if there is a direct path to any negative applications, the authors should point it out. For example, it is legitimate to point out that an improvement in the quality of generative models could be used to generate deepfakes for disinformation. On the other hand, it is not needed to point out that a generic algorithm for optimizing neural networks could enable people to train models that generate Deepfakes faster.
- The authors should consider possible harms that could arise when the technology is being used as intended and functioning correctly, harms that could arise when the technology is being used as intended but gives incorrect results, and harms following from (intentional or unintentional) misuse of the technology.

- If there are negative societal impacts, the authors could also discuss possible mitigation strategies (e.g., gated release of models, providing defenses in addition to attacks, mechanisms for monitoring misuse, mechanisms to monitor how a system learns from feedback over time, improving the efficiency and accessibility of ML).

11. **Safeguards**

Question: Does the paper describe safeguards that have been put in place for responsible release of data or models that have a high risk for misuse (e.g., pretrained language models, image generators, or scraped datasets)?

Answer: [NA]

Justification: Our work contributes to improving the correctness of hardware systems. The benchmark designs used in our evaluation are extensions of a prior benchmark suite and consist of standard hardware modules drawn from well-known literature. These designs are in the public domain and do not pose any risk of misuse.

Guidelines:

- The answer NA means that the paper poses no such risks.
- Released models that have a high risk for misuse or dual-use should be released with necessary safeguards to allow for controlled use of the model, for example by requiring that users adhere to usage guidelines or restrictions to access the model or implementing safety filters.
- Datasets that have been scraped from the Internet could pose safety risks. The authors should describe how they avoided releasing unsafe images.
- We recognize that providing effective safeguards is challenging, and many papers do not require this, but we encourage authors to take this into account and make a best faith effort.

12. **Licenses for existing assets**

Question: Are the creators or original owners of assets (e.g., code, data, models), used in the paper, properly credited and are the license and terms of use explicitly mentioned and properly respected?

Answer: [Yes]

Justification: We include the licenses for all tools stated in section 5, and provide appropriate citations for each.

Guidelines:

- The answer NA means that the paper does not use existing assets.
- The authors should cite the original paper that produced the code package or dataset.
- The authors should state which version of the asset is used and, if possible, include a URL.
- The name of the license (e.g., CC-BY 4.0) should be included for each asset.
- For scraped data from a particular source (e.g., website), the copyright and terms of service of that source should be provided.
- If assets are released, the license, copyright information, and terms of use in the package should be provided. For popular datasets, `paperswithcode.com/datasets` has curated licenses for some datasets. Their licensing guide can help determine the license of a dataset.
- For existing datasets that are re-packaged, both the original license and the license of the derived asset (if it has changed) should be provided.
- If this information is not available online, the authors are encouraged to reach out to the asset's creators.

13. **New assets**

Question: Are new assets introduced in the paper well documented and is the documentation provided alongside the assets?

Answer: [Yes]

Justification: We include the source code of our prototype and the extended benchmark suite as supplementary material to this paper, released under the MIT License.

Guidelines:

- The answer NA means that the paper does not release new assets.
- Researchers should communicate the details of the dataset/code/model as part of their submissions via structured templates. This includes details about training, license, limitations, etc.
- The paper should discuss whether and how consent was obtained from people whose asset is used.
- At submission time, remember to anonymize your assets (if applicable). You can either create an anonymized URL or include an anonymized zip file.

14. **Crowdsourcing and research with human subjects**

Question: For crowdsourcing experiments and research with human subjects, does the paper include the full text of instructions given to participants and screenshots, if applicable, as well as details about compensation (if any)?

Answer: [NA]

Justification: This paper does not involve crowdsourcing or research with human subjects.

Guidelines:

- The answer NA means that the paper does not involve crowdsourcing nor research with human subjects.
- Including this information in the supplemental material is fine, but if the main contribution of the paper involves human subjects, then as much detail as possible should be included in the main paper.
- According to the NeurIPS Code of Ethics, workers involved in data collection, curation, or other labor should be paid at least the minimum wage in the country of the data collector.

15. **Institutional review board (IRB) approvals or equivalent for research with human subjects**

Question: Does the paper describe potential risks incurred by study participants, whether such risks were disclosed to the subjects, and whether Institutional Review Board (IRB) approvals (or an equivalent approval/review based on the requirements of your country or institution) were obtained?

Answer: [NA]

Justification: This paper does not involve crowdsourcing or research with human subjects.

Guidelines:

- The answer NA means that the paper does not involve crowdsourcing nor research with human subjects.
- Depending on the country in which research is conducted, IRB approval (or equivalent) may be required for any human subjects research. If you obtained IRB approval, you should clearly state this in the paper.
- We recognize that the procedures for this may vary significantly between institutions and locations, and we expect authors to adhere to the NeurIPS Code of Ethics and the guidelines for their institution.
- For initial submissions, do not include any information that would break anonymity (if applicable), such as the institution conducting the review.

16. **Declaration of LLM usage**

Question: Does the paper describe the usage of LLMs if it is an important, original, or non-standard component of the core methods in this research? Note that if the LLM is used only for writing, editing, or formatting purposes and does not impact the core methodology, scientific rigorousness, or originality of the research, declaration is not required.

Answer: [NA]

Justification: The core method development in this research does not involve large language models (LLMs) as any important, original, or non-standard components.

Guidelines:

- The answer NA means that the core method development in this research does not involve LLMs as any important, original, or non-standard components.
- Please refer to our LLM policy (`https://neurips.cc/Conferences/2025/LLM`) for what should or should not be described.

