# OpenReview forum: "Let a Neural Network be Your Invariant"
_NeurIPS.cc/2025/Conference — NeurIPS 2025 poster_

### Official Review · Reviewer_3nK8 · 2025-06-07

**Clarity:** 4
**Significance:** 4
**Originality:** 3
**Rating:** 5
**Confidence:** 4

**Summary:**

This paper introduces a novel extension to neural model checking by enabling neural networks to serve simultaneously as inductive invariants (for safety verification) and ranking functions (for liveness verification). The authors propose a unified neural certificate architecture—called a partially-ranking function—that classifies reachable states (safety) and ranks their progress toward satisfying liveness properties. Unlike prior work that uses gradient descent, this approach leverages constraint solvers (e.g., SMT, MILP) for training, improving both correctness guarantees and computational efficiency.
The authors demonstrate the practicality and effectiveness of the method through extensive experiments on 634 benchmarks from SystemVerilog designs, showing up to big speedups over both traditional symbolic model checkers and previous neural approaches, especially for combined safety-liveness verification tasks.

**Questions:**

Can the authors elaborate on the frequency and nature of solver errors encountered during experiments? Would fallback strategies be practical or recommended?

Do the authors foresee limitations in applying this to software model checking, such as pointer-based memory or concurrency?

The architecture is piecewise-linear and intentionally simple for SMT compatibility. How expressive is it for more complex invariants/rankings?

In cases where the neural certificate fails to converge, how does one determine whether the issue is with architecture capacity, undersampling, or unsatisfiability of the spec?

**Ethical Concerns:**

["NO or VERY MINOR ethics concerns only"]

**Final Justification:**

I am fine with the authors' answer and will maintain my score.

**Limitations:**

Future work could strengthen solver diagnostics and scalability to more diverse systems.

**Paper Formatting Concerns:**

N.A.

**Quality:**

3

**Strengths And Weaknesses:**

Strengths

- The paper presents a robust and formally grounded method that advances the frontier of neural formal verification. The constraint-based training of partially-ranking functions is innovative and mathematically sound.
- The paper is well-structured and includes motivating examples, mathematical formulation, and diagrams that effectively communicate the ideas.
- The approach addresses a critical gap in existing neural model checking, namely the inability to reason about safety properties, and brings it to parity with classical model checking tools.
- The unification of safety and liveness certificates into a single neural network and the replacement of gradient descent with SMT/MILP solvers are novel contributions.

Weaknesses

- Although the authors scale to realistic hardware designs, it remains unclear how well the method generalizes to larger or more complex systems
- The method depends heavily on solver behavior and issues like solver unsoundness or numerical instability are not deeply analyzed

---

> ### Author Rebuttal · Authors · 2025-07-30
>
> Q1. We thank the reviewer for suggesting a failure analysis. After re‑running the pure‑liveness suite, we found the failure causes to be as follows: 18 % were numerical mismatches between Gurobi’s solution and our NumPy sanity check; 39 % gave “model infeasible’’ reports for every architecture explored by our‑auto (we consider between zero to five hidden neurons); 21 % were segmentation faults; 7 % exited complaining that the MILP exceeds our size‑limited licence; and 14 % were time‑outs. We will report the failure analysis for the entire task suite in subsequent versions.
>
>
> In total, across all benchmarks, Gurobi fails 98 times out of 634 tasks. We note that our method already includes a fallback strategy. A simple two‑solver portfolio—launching a sound solver in parallel (e.g., Z3) and considering the first solver to finish the entire task—rescues 59 of these 98 cases within the time cap with Z3. Fig. 4(c) shows the improvement possible from a four-solver portfolio through the Best curve. We will incorporate these numbers and emphasise the portfolio fallback strategy in the revision.
>
> Q2. Our method learns ranking functions and inductive invariants from data, relying on knowledge of the number and types of allocated variables. This assumption is specific to hardware and, in many cases, to embedded software. Generalising our approach to software with dynamic memory allocation as well as concurrency would require the use of compositional proof rules. This challenge applies to all certificate synthesis methods—not just neural approaches—and represents a compelling direction for future work.
>
> Q3. As correctly noted, our architecture is piecewise-linear—a design choice that ensures compatibility with linear theories within SMT solvers. In theory, our architecture could represent any invariant or ranking function over a finite state space by effectively encoding a large look-up table. However, this is clearly impractical except for trivial cases. The more relevant and open question is what can be expressed succinctly—that is, with a small number of neurons. This remains an intriguing direction for future research on neural certificates.
>
> Q4.Our method distinguishes undersampling from the other two failure cases. Specifically, when the learner succeeds but the checker returns a counter-example, we add it to the training set for the next learn-check iteration, mitigating undersampling. Otherwise, if the learner fails—and assuming the learner is complete—we can infer that either the network architecture lacks sufficient capacity or the specification is unsatisfiable. In this case, our algorithm increases the network size, and this scenario arises in our experiments. In principle, we could eventually distinguish between capacity limitations and unsatisfiability. In the worst case, we would enumerate all possible states (which are finite) and construct a network large enough to function as a look-up table, represented as a large piecewise-linear function. However, distinguishing capacity limitations and unsatisfiability with this approach is only practically feasible for trivial verification tasks before hitting the timeout.

---

> > ### Comment · Reviewer_3nK8 · 2025-08-05
> >
> > Thanks for the detailed response and clarifications. I will keep my score.

---

### Official Review · Reviewer_MKHH · 2025-07-01

**Clarity:** 3
**Significance:** 3
**Originality:** 3
**Rating:** 5
**Confidence:** 4

**Summary:**

This paper presents a novel approach to model checking of reactive systems against full linear temporal logic (LTL) specifications. Unlike prior neural methods, which were limited to proving liveness properties via learned ranking functions, this work introduces a unified neural certificate that simultaneously encodes inductive invariants (for safety) and ranking functions (for liveness).

The key innovation lies in training these neural certificates using constraint solving, rather than gradient descent, and verifying their correctness via a single symbolic check. The authors demonstrate that their method is sound, scalable for many practical tasks, and significantly outperforms both prior neural model checkers and established symbolic model checking tools on an extended benchmark suite.

**Questions:**

- How does the method handle specifications where the Büchi automaton for ¬Φ is very large or nondeterministic? Can the approach scale symbolically in that case?
- Would your method benefit from hybrid approaches (e.g., warm-starting with gradient descent and refining with constraint solving)?
- How do you plan to scale your method to train large networks specifically how you use the solver efficiently?

**Ethical Concerns:**

["NO or VERY MINOR ethics concerns only"]

**Final Justification:**

Thanks for the rebuttal. I will keep my score as is.

**Limitations:**

yes

**Quality:**

3

**Strengths And Weaknesses:**

Strengths
-----------
 - Clear Technical Innovation: The central idea—joint learning of safety and liveness proofs using a unified neural network—is novel and well-motivated. Prior work addressed liveness only; this paper closes that critical gap.
 - Soundness Preserved: Despite using machine learning components, the approach guarantees formal soundness through symbolic verification. The combination of logic-based training and model checking is elegant and rigorous.
 - Strong Empirical Performance: On a comprehensive 634-task benchmark, the method outperforms five decades of symbolic model checking techniques in many categories, especially for properties combining safety and liveness. Reported speedups (up to 10⁴×) are significant.

Weaknesses
---------------
 - Scalability Limits of Constraint Solvers: Although the paper reports excellent performance on the benchmark suite, it does not address how the constraint-solver-based training will scale to systems with large state spaces or complex invariants. Solvers tend to struggle with large, nonlinear constraint encodings—this bottleneck should be discussed more explicitly.
- Baseline selection: The evaluation is strong, but lacks a direct comparison with the most recent symbolic checkers such as rIC3 (https://arxiv.org/html/2502.13605v1). It's unclear whether the proposed method would retain its performance advantage in those cases.
 - Assumes Büchi Automata Are Given: The approach assumes that the violation automaton for ¬Φ (a nondeterministic Büchi automaton) is already constructed. This step can be a significant practical hurdle, especially for real-world LTL specifications.

---

> ### Author Rebuttal · Authors · 2025-07-30
>
> Q1. Our method handles specifications whose corresponding automaton is non-deterministic by allowing counterexample transitions to be generated along all possible branches of the automaton. Regarding the question of whether we experimented with large specifications, we note that in most practical cases, industrial specifications are composed of lists of relatively small temporal logic obligations; see, for example, "Toward Liveness Proofs at Scale" (CAV 2024). Our work aligns with and captures this standard scenario.
>
> Q2. Our method would not benefit from warm-starting the constraint solver using gradient descent, for the simple reason that the solvers we employ neither require nor expect an initial value. Assessing whether our approach could benefit from warm-starting would require integrating alternative solvers—a direction that falls outside the scope of this manuscript but is indeed a very good suggestion for future work.
>
> Q3. Our method indeed benefits from using compact neural networks, and our experiments show that such models are sufficient in practice on a standard benchmark. In fact, our approach explicitly searches for the smallest network possible for a given benchmark, since (as the reviewer points out) constraint solvers tend to struggle with very large problem instances. In this paper, we argue that the key to scalable verification with neural certificates lies in using small networks, rather than scaling verification algorithms to handle large networks. That said, we acknowledge that for other verification tasks, scaling to larger networks might prove more effective. Exploring this trade-off is an important direction for future research on neural certificates.
>
> Finally, we thank the reviewer for suggesting rIC3. We have evaluated the publicly available “16‑threads Portfolio rIC3” implementation (github.com/gipsyh/rIC3). On tasks that are easy for most tools, rIC3 outperforms other symbolic approaches, consistent with the authors’ own evaluation. However, on harder instances, the performance gap narrows, and rIC3 aligns with other symbolic tools; in fact, for 40% of tasks, nuXmv outperforms rIC3. Nevertheless, overall—and especially when viewed on log-scale cactus plots—our method remains the clear winner outside of pure safety verification. All statistics comparing our tool against the best-performing tools run in parallel change by at most 3 percentage points, leaving our conclusions unchanged. As NeurIPS'25 rebuttal policy precludes the inclusion of new figures, we will provide the updated experiments only in the revised version of the manuscript.

---

> > ### Comment · Reviewer_MKHH · 2025-08-04
> > **Reviewer response**
> >
> > Thanks for the detailed response and clarifications! I will like to keep my score.

---

### Official Review · Reviewer_Z4tK · 2025-07-01

**Clarity:** 2
**Significance:** 2
**Originality:** 3
**Rating:** 3
**Confidence:** 4

**Summary:**

The paper presents a new approach for proving safety and liveness of reactive systems, where the proofs are represented using neural networks. The results build on recent results, including [56,57], which address temporal model checking and liveness verification, respectively. In this paper, reactive systems can be construed as state-transition systems, which admit variables and that transition relations, set of initial states, etc. are described symbolically (for instance, by means of SMT formulas). The proof techniques adopted in the paper are in fact standard in the verification literature. To prove safety, one establishes a program invariant. To prove liveness, one provides a ranking function. The authors prove the two properties at the same time using the notion of "partially" ranking function: the proof rules (5) - (7). In this paper, the authors propose to represent such a partially ranking function as a neural network (i.e. a "neural certificate"). Neural certificates are not new, as the authors also describe in Section 5, but the paper is the first time when they are employed for partially ranking functions. The authors then propose a new learning algorithm for inferring neural certificates, which are curiously not based on standard ML machinery, but based on SMT solvers. In fact, the SMT-based learning algorithm is reminiscent of CEGAR-based formula synthesis that exploits constraint solvers. The connection between neural certificates and SMT solvers lies in the well-known fact that neural networks (with ReLU activation functions) can be translated into MILP (Mixed Integer Linear Programming), which can in turn be solved by most SMT-solvers.

In the implementation and experiments, the work assumes a SystemVerilog input, which consists of a program and a property to verify. The implementation combines many existing tools including SPOT to convert an LTL formula into an automaton, EBMC to convert SystemVerilog program into SMT over bitvectors, bitwuzla to solve SMT over bitvectors, and Gurobi to solve MILP constraints. Experiments are conducted on an extension of prior benchmarks from [57], which includes both safety and liveness properties. In total, this is a 634 task suite. The authors report highly encouraging experimental results, in particular outperforming existing symbolic approaches on safety-liveness properties.

**Questions:**

Please respond to what I mentioned above seem to be weaknesses of the paper.

**Ethical Concerns:**

["NO or VERY MINOR ethics concerns only"]

**Final Justification:**

During the rebuttal, raised limitations were acknowledged by the authors. Since such neural certificate could still be represented efficiently by SMT formulas, it's still unclear to me the advantage of the neural representation. For these reasons, I've decided to keep my original score.

**Limitations:**

Yes

**Quality:**

3

**Strengths And Weaknesses:**

Strengths:
- the paper attacks an important problem in verification
- the use of neural certificates for the first time for safety-liveness properties
- CEGAR learning algorithm to infer neural certificates that exploits SMT solvers
- positive experimental results showing competitiveness of the technique

Weaknesses:
- Benefits of neural networks are unclear, upon further reflection.
- Restrictions in the allowed programs by the proposed approach are unclear.

Let me elaborate more on what I perceive as weaknesses of the paper; the authors should address these during the rebuttal. Firstly, the main benefit of neural networks is the existence of backpropagation algorithms that use gradient descent. This is not used in the paper, which opted to "train" the neural networks using constraint solvers instead of gradient descent. The authors position the contribution of the paper as "using neural networks as proofs". Since we know that neural networks can be seen as MILP formulas, it seems that the entire approach can simply be seen as CEGAR-based formula synthesis (by using SMT-solvers as oracles) and one can dispense of neural networks from their approach. This raises the questions of relevance of the paper to an ML conference.

Secondly, the authors were not very clear in the definition in the preliminaries (Section 2) of what kind of variables are allowed in the programs, and what kind of data types and operations are allowed. There are hints on these in Section 3 and Section 4, based on results and tools that are used to translate from SystemVerilog to neural networks. That is, the programs use bitvectors, but for example no arrays, strings, etc. It would be good if the authors provide more details on the limitation of their approach. What type of programs can your approach handle in theory?

Finally, the paper did a reasonable job in explaining basic concepts in formal verification for an ML conference. That said, there are a few definitions that the authors forgot to define (e.g. MILP).

---

> ### Author Rebuttal · Authors · 2025-07-30
>
> The reviewer highlights two central aspects of our work, which we will emphasise in the revised version.
>
> First, it is indeed correct that one of the main advantages of neural networks is their compatibility with gradient descent—an aspect we do not exploit in this work. However, this is not the only benefit neural networks offer. They are powerful representations of non-linear functions, independent of the training algorithm used. Our work hands down leverages this representational power. We have shown that our architecture is an appropriate representation for invariant and ranking functions on a standard benchmark. And most importantly, our ablation study has shown that linear functions are insufficient.  It is the use of neural networks to effectively represent non-linear invariant and ranking functions that allows our method to dominate the state of the art.
>
> Second, the reviewer is correct in noting that our method is limited to systems where the number and size of variables are known a priori. Our method does not directly extend to arrays and strings of parametric length, or to dynamic data structures, being specifically tailored to hardware and embedded software that does not rely on stack or heap memory. Extending it to general-purpose software presents significant challenges and represents an important direction for future research.
>
> Finally, we appreciate pointing out the lack of certain definitions, which we will expand upon in the revised version.

---

> > ### Comment · Reviewer_Z4tK · 2025-08-04
> >
> > Thank you for the clarification. In view of the pinpointed limitations, with which the authors agreed, I maintain my score.

---

> > > ### Author Response · Authors · 2025-08-05
> > >
> > > We thank the reviewer for acknowledging our response. We concur with the points raised, but we also emphasize that these are features and not inherent limitations of our work. We demonstrated that neural networks are powerful representations for both inductive invariants and ranking functions. While related work on neural model checking (NeurIPS '24) has already employed gradient descent for liveness verification (with which we provide a direct comparison), our paper further demonstrates that solver-based techniques are faster. Our technology targets the standard hardware model checking problem setting and supports full SystemVerilog that is synthesisable as circuit, which does not have dynamic data types.

---

### Official Review · Reviewer_jK1G · 2025-07-08

**Clarity:** 3
**Significance:** 3
**Originality:** 3
**Rating:** 5
**Confidence:** 3

**Summary:**

This paper addresses the problem of verifying verilog circuits by conjecturing
proof certificates using a neural network. Prior work has shown acceleration of
liveness properties by having a neural network conjecture a measure of progress
(a ranking function.) The key contribution of this work is to also predict
a safety invariants. These are then fed into a formal verification pipeline
to assert correctness. The method of learning the neural network departs
from other systems in that it uses a constraint solver, e.g. SMT, to
set the neural network weights or prove that there exists no neural network
that satisfies a set of constraints. This enables smaller networks by using
linearly searching the space of neural networks. Finally, they show that
this method accelerates or is competitive with existing non-neural solvers.

**Questions:**

1. How was the architecture family selected?

2. What are the limitations on the neural networks that can be used.

3. Do you see a path towards learning to generalize to multi-task settings? i.e., taking the verliog as input and learning to interpolate over verliog programs?

**Ethical Concerns:**

["NO or VERY MINOR ethics concerns only"]

**Final Justification:**

I've maintained my score. The main criticisms raised through the review process seem to be (i) the lack of expressiveness of the target class of NN (ii) the learning procedure (iii) the subtle restriction of verilog programs

I think (i) and (ii) while restrictions, point more to future directions and opportunities than failings of the paper. (iii) is indeed a failing and I would encourage the authors to better address, e.g., in an appendix, but it is not severe enough for me to lower my score.

**Limitations:**

yes

**Quality:**

3

**Strengths And Weaknesses:**

## Strength

1. The domain of verilog verification is very important -- both historically
   and particularly given the explosion in bespoke hardware designs for AI
   acceleration.

2. The provided speed gains seem non-trivial, particularly in a portfolio suite.

3. The method produces much smaller networks, and actually provably the smallest
   in the class explored, than a naive overparameterized model.

## Weakness


1. The largest weakest from my reading, and one that I'd appreciate more exploration
   in a future draft, is the class of Neural Networks supported by both the synthesis
   and verification queries. I particular, my understanding is the class of
   non-linearitie is limited by the theories (efficiently) covered by the SMT
   solvers.

2. Relatedly, while interesting the counter example loop -- in the formal methods
   literature an instance of CEGIS? -- implicitly relies on refuting smaller neural
   architectures having sufficient capacity. This seems to work well in these benchmarks,
   but may catastrophically fail for larger problems.

3. As a minor criticism, I disagree with the discussion around quantization --
   namely that the quantization need not be learned. In an information
   theoretic sense, the optimal quantization is necessarily dependent on the
   input distribution and downstream arithmetic circuit. Fixing the quantization
   a-priori necessarily loses expressivity. That said, I think the advantages
   for fitting into SMT theories might outweigh the downsides.

---

> ### Author Rebuttal · Authors · 2025-07-30
>
> The reviewer correctly notes that our work is restricted to neural networks with rectified activation functions—a choice that is common in both machine learning and verification literature. In our context, this design is specifically motivated by the need to encode networks into efficient (linear) theories of SMT solvers. We will expand on this important point and provide the details of our symbolic encoding in the revised version.
>
> We confirm that our approach is indeed an instance of a CEGIS loop. As with most CEGIS-based best-effort methods, there remains scope for tackling more challenging problems—an open direction for future research.
>
> The question of whether the appropriate quantisation should be learned is an important one, relevant to all approaches that employ quantised neural networks. In our case, the learning procedure produces networks with integer coefficients, reducing the quantisation to simply estimate the maximum range (and not the minimum precision), which is a significantly simpler quantisation task than in the general setting—and notably simpler than in cognate work such as neural model checking (NeurIPS’24).
>
> Q1. Our architecture family was specifically designed to balance trainability and verifiability, leveraging efficient SMT theories—namely, Linear Integer Arithmetic (LIA/MILP) for training and Bit-Vector (BV) theory for verification. As noted above, we will provide further elaboration on this point in the revised version.
>
> Q2. As such, while the general approach of training neural certificates as invariants  applies in principle to every feed-forward architecture, the algorithm based on SMT that we presented in this paper benefits from network families that have rectified activation functions (sign and ReLU) and discrete values domains.
>
> Q3. Our approach, which learns from system executions, is designed for single-task, unsupervised settings. Whether it can be generalised to support transfer learning across tasks remains an open question for future work—one that would require more than a straightforward extension of the work presented in this paper.

---

> > ### Comment · Reviewer_jK1G · 2025-08-07
> >
> > Thank you for your comments. I will keep my score as I think my concerns/questions were sufficiently addressed and I don't have the same issue about gradients as Z4tK.
> >
> > I do agree with Z4tK though that the limits on the class of programs that can be checked would be a nice addition. If this were a CAV paper or similar, I would more strongly assert this, but I think for Neurips even having a discussion in an appendix is sufficient.

---

### Decision · Program_Chairs · 2025-09-17

**Decision:**

Accept (poster)

**Comment:**

This work proposes a neural network certificate for proving the safety and liveness of reactive systems, where the neural network aims to learn a partial ranking function. The proof techniques (e.g., invariants and ranking functions) themselves are classical, but the work is the first to demonstrate that the partial ranking functions can be represented and learned via a neural network for an LTL specification. The evaluation is based on practical SystemVerilog verification problems, and a test suites with a large number of instances were demonstrated with promising results. Overall, the contribution of this work is novel and above the bar for NeurIPS publication.

One limitation of this work is the use of SMT/MILP solver to deal with neural network certificates, which inherently limits the neural network size and complexity. I suggest that the authors look into the tools developed by the neural network verification community, such as the tools proposed in VNN-COMP (https://arxiv.org/abs/2412.19985), which should provide significantly better scalability to further enhance the practicality and scalability of this work.

The use of a constraint programming solver in the learning approach is potentially another limitation, although it can work better on smaller-scale problems. Recent works in neural certificates, such as https://arxiv.org/pdf/2404.07956 and https://arxiv.org/pdf/2506.01356 have moved away from constraint programming and SMT for trading off scalability. I hope the authors can consider these factors in their future work.